# LapFlow: Laplacian Multi-scale Flow Matching for Generative Modeling

**Zelin Zhao & Petr Molodyk & Haotian Xue & Yongxin Chen** *
Georgia Institute of Technology
Atlanta, GA 30332, USA
`{zelin,pmolodyk3,htxue.ai,yongchen}@gatech.edu`

## Abstract

In this paper, we present Laplacian multiscale flow matching (LapFlow), a novel framework that enhances flow matching by leveraging multi-scale representations for image generative modeling. Our approach decomposes images into Laplacian pyramid residuals and processes different scales in parallel through a mixture-of-transformers (MoT) architecture with causal attention mechanisms. Unlike previous cascaded approaches that require explicit renoising between scales, our model generates multi-scale representations in parallel, eliminating the need for bridging processes. The proposed multi-scale architecture not only improves generation quality but also accelerates the sampling process and promotes scaling flow matching methods. Through extensive experimentation on CelebA-HQ and ImageNet, we demonstrate that our method achieves superior sample quality with fewer GFLOPs and faster inference compared to single-scale and multi-scale flow matching baselines. The proposed model scales effectively to high-resolution generation (up to 1024Œ1024) while maintaining lower computational overhead.

## 1 Introduction

Generative modeling has seen remarkable progress in recent years, with diffusion models and flow matching approaches achieving state-of-the-art results in image synthesis (Dhariwal and Nichol, 2021; Lipman et al., 2023; Peebles and Xie, 2023). These models have demonstrated impressive capabilities in generating realistic, high-quality, and diverse samples across various domains. However, as the demand for higher resolution and more complex content increases, scalability remains a significant practical challenge (Tian et al., 2024; Jin et al., 2025; Atzmon et al., 2024). As these methods typically generate the entire image at full resolution, they require substantial computational resources during training and inference phases. This motivates alternative approaches that maintain generation quality while potentially improving sampling efficiency and reducing inference overheads.

Multi-scale generation (Lai et al., 2017; Jin et al., 2025; Atzmon et al., 2024; Tian et al., 2024; Ho et al., 2022; Batzolis et al., 2022) has emerged as a promising direction to address these scalability challenges. Existing multi-scale methods in both diffusion and flow matching paradigms, such as Cascaded Diffusion Models (Ho et al., 2022), EdifyImage (Atzmon et al., 2024), and Pyramidal Flow (Jin et al., 2025), have demonstrated improved efficiency by generating images progressively from lower to higher resolutions. However, each of these approaches faces challenges that have limited their adoption compared to popular single-scale methods like DiT (Peebles and Xie, 2023). Cascaded Diffusion Models (Ho et al., 2022) require training and maintaining separate networks for each resolution level, increasing implementation complexity. EdifyImage (Atzmon et al., 2024), while effective, operates directly in pixel space rather than latent space, resulting in significantly slower inference compared to latent-based methods. Pyramidal Flow (Jin et al., 2025) has demonstrated impressive results when fine-tuning DiTs for video generation, where they initialize their MM-DiT model from SD3 Medium (Esser et al., 2024) and fine-tune it afterwards, but its effectiveness when trained from scratch for image generation tasks remains less thoroughly explored in the literature.

In this paper, we introduce *Laplacian Multi-scale Flow Matching* (`LapFlow`), enabling parallel modeling of multi-scale representations. As shown in Figure 1, the proposed model progressively

---

*The code is available at https://github.com/sjtuytc/gen.

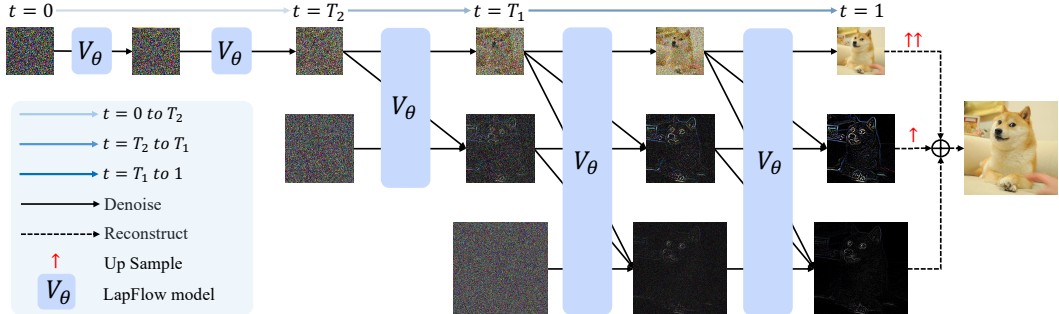

Figure 1: **Multi-scale generation process of our model**. The proposed model follows a coarse-to-fine generation strategy across scales in a Laplacian pyramid. This figure demonstrates a three-level version of ours, where $T_2$, $T_1$ are two critical points defining three sampling segments for three scales. Starting from a random noise at $t = 0$, our model first denoises the coarsest scale until $t = T_2$, then progressively conditions finer scales on completed coarser scales ($t = T_2$ to $T_1$ and $t = T_1$ to $1$). This causal structure ensures coherent image generation by maintaining hierarchical dependencies across scales, ultimately producing high-fidelity samples with both global consistency and fine details.

generates Laplacian residuals through a unified model, and then reconstructs the full image through a hierarchical combination of these residuals following the Laplacian pyramid reconstruction process. Our approach enables parallel generation of multiple scales through a unified *mixture-of-transformers* (MoT (Liang et al., 2024)) model with scale-specific modeling and shared-weight global attention. The model further employs causal attention mechanisms that enforce a natural information flow from lower to higher resolution scales, ensuring that finer details are coherently conditioned on broader structural elements while maintaining the hierarchical integrity of the image representation.

Through extensive experimentation on both CelebA-HQ (Karras et al., 2018) and ImageNet (Deng et al., 2009), we demonstrate that our approach achieves superior sample quality compared to existing single-scale and multi-scale flow matching methods while requiring fewer compute (GFLOPs) during sampling. On CelebA-HQ, our method achieves an FID of $3.53$ at $256×256$ resolution (compared to $5.26$ for LFM (Dao et al., 2023)), while our method keeps a strong performance scaling up to $1024 \times 1024$. Comprehensive ablation studies on CelebA-HQ demonstrate the effectiveness of our key design choices, including the MoT architecture, causal masking, and noise scheduler, etc. For class-conditional generation on ImageNet, our approach outperforms both single-scale and multi-scale flow matching methods while maintaining lower computational requirements.

**Our contributions are summarized as follows:**

1. We present a multi-scale flow matching framework that decomposes images into Laplacian pyramid representations, enabling joint modeling of different scale components.

2. We introduce a specialized mixture-of-transformers (MoT) architecture with causal attention mechanisms that processes multiple scales simultaneously, significantly reducing inference compute while enforcing natural information flow between resolution levels. Through a time-weighted complexity analysis, we show that the effective attention cost of our progressive multi-scale design is theoretically lower than that of DiT (Peebles and Xie, 2023).

3. We develop a progressive training strategy that optimizes different scales across distinct time ranges, allocating computational resources according to each scale's contribution.

## 2 RELATED WORK

**Diffusion and flow matching.** Diffusion models (Peebles and Xie, 2023; Ho et al., 2020) have emerged as a dominant approach for generative modeling, achieving state-of-the-art results across various domains through iterative denoising processes. Despite their success, these models often require a large number of function evaluations during sampling, leading to computational inefficiency. Flow matching (Lipman et al., 2023; Albergo and Vanden-Eijnden, 2022; Liu et al., 2022) is a simple yet effective framework for generative modeling, demonstrating strong performance across multiple

domains, particularly in image generation (Ma et al., 2024; Esser et al., 2024; Liu et al., 2022; Dao et al., 2023) and video generation (Jin et al., 2025; Google DeepMind, 2025). Instead of using noise or clean samples as learning objectives, it learns a velocity field that defines a deterministic transformation from the prior to the target distribution.

**Multi-scale generation.** Multi-scale approaches have a rich history in generative modeling, beginning with LapGAN (Lai et al., 2017), which demonstrated the effectiveness of progressively generating high-resolution images from lower-resolution ones using GANs. This hierarchical concept was later adapted to diffusion models with cascaded architectures (Ho et al., 2022; Saharia et al., 2022; Zhang et al., 2023), which employ sequences of separate models to generate images of increasing resolution, each conditioned on lower-resolution outputs. Later on, Non-Uniform Diffusion Models (Batzolis et al., 2022), Relay Diffusion (Teng et al., 2024) and Pyramidal Flow (Jin et al., 2025) utilize renoising operations to bridge between resolution levels in a full generation cycle, while EdifyImage (Atzmon et al., 2024) attenuates different frequency components at varying rates. However, these approaches typically require separate models or complex bridging mechanisms between scales, ignoring the use of causal relationships between scales (Tian et al., 2024). In contrast, our `LapFlow` framework employs a single mixture of transformers model that processes all scales simultaneously through causal attention mechanisms, eliminating the need for explicit bridging while allowing for more efficient generation.

**Parameter-efficient architectures and auto-regressive generative models.** Parameter efficiency has become increasingly important in generative modeling as model sizes continue to grow. Mixture-of-Experts (MoE) approaches (Shazeer et al., 2017; Fedus et al., 2022) have emerged as a powerful paradigm for conditionally activating only relevant parameters, substantially improving computational efficiency. These concepts have been adapted to vision tasks through Mixture-of-Transformers (MoT) (Liang et al., 2024) architectures that enable specialized processing pathways. Concurrent developments in auto-regressive generative models (Tian et al., 2024; Sun et al., 2024) have demonstrated the effectiveness of causal modeling. Our approach leverages flow matching with ODE-based parallel sampling, fundamentally differentiating it from auto-regressive methods that generate content sequentially and suffer from inherent parallelization constraints.

## 3 BACKGROUND

**Flow matching formulations.** Flow matching (Lipman et al., 2023) learns a time-dependent vector field $\mathbf{u}_t$ that smoothly interpolates between a prior distribution $p_0$ and a data distribution $q$. We use $\mathbf{x}_t$ to represent the evolving data sample over time. Flow matching directly optimizes a regression objective $\mathbb{E}_{t,q(\mathbf{x}_1),p_t(\mathbf{x}|\mathbf{x}_1)} \|\mathbf{v}_t(\mathbf{x}) - \mathbf{u}_t(\mathbf{x}|\mathbf{x}_1)\|^2$, minimizing the discrepancy between the learned vector field $\mathbf{v}_t(\mathbf{x})$ and the conditional velocity of ground truth $\mathbf{u}_t(\mathbf{x}|\mathbf{x}_1)$, called the Conditional Flow Matching (CFM) objective. Training a flow-matching model requires defining a valid interpolation path between $p_0$ and $q$. Let $\mathbf{x}_0 \sim p_0(\mathbf{x})$ be a noise variable, and $\mathbf{x}_1 \sim q(\mathbf{x})$ be a data sample. We can derive the noisy data $\mathbf{x}_t$ and the corresponding velocity field $\mathbf{u}_t$ as $\mathbf{x}_t = \alpha_t \mathbf{x}_1 + \sigma_t \mathbf{x}_0$ and $\mathbf{u}_t(\mathbf{x}_t|\mathbf{x}_1) = \dot{\alpha}_t \mathbf{x}_1 + \dot{\sigma}_t \mathbf{x}_0$, where $\alpha_t$ and $\sigma_t$ are pre-defined coefficients for noise scheduling, while $\dot{\alpha}$ and $\dot{\sigma}$ are their temporal derivatives respectively. A simple yet effective choice (Lipman et al., 2023; Dao et al., 2023) of $\alpha_t$ and $\sigma_t$ is linear interpolation: $\alpha_t = t, \sigma_t = 1 - t$. Another popular probability path is the generalized variance preserving (GVP) (Ma et al., 2024): $\alpha_t = \sin\left(\frac{1}{2}\pi t\right), \sigma_t = \cos\left(\frac{1}{2}\pi t\right)$. We choose to model flows in latent space (Dao et al., 2023), aiming to reduce computational complexity and improve sample quality.

**Latent flow matching.** While some flow matching methods attempt to directly model in pixel space (Lipman et al., 2023), LFM (Dao et al., 2023) and SiT (Ma et al., 2024) propose to model in the latent space through a VAE, aiming to reduce computational complexity and improve sample quality.

**Laplacian decomposition.** Denote the unknown data distribution (the image distribution) by $q(\mathbf{x}_1)$. We define Up as the nearest neighbor upsampling operation and Down as the average pooling operation, both with a scaling factor of two for upsampling and downsampling, respectively. Given a data $\mathbf{x}_1 \sim q$, we decompose it into three residuals via the Laplacian decomposition (we formulate

---

**Algorithm 1** `LapFlow` Training

---

1: **Input:** image dataset $D$, number of epochs $M$, critical time points $0 \triangleq T_3 < T_2 < T_1 < 1$
2: Initialize weights $\theta$ of the `LapFlow` model $V_\theta$
3: **for** epoch = 1 to $M$ **do**
4:     **for** each image $\mathbf{x}_1$ in $D$ **do**
5:         Sample a random noise image $\mathbf{x}_0$ and generate Laplacian pyramid $\{\mathbf{x}_0^{(0)}, \mathbf{x}_0^{(1)}, \mathbf{x}_0^{(2)}\}$
6:         Sample stage $s \sim \mathcal{U}\{0, 1, 2\}$, and sample time $t \sim \mathcal{U}[T_{s+1}, 1]$
7:         Compute noisy image through Equation (3) for $k \geq s$ and $k \in \{0, 1, 2\}$
8:         Compute velocity target through Equation (5) for $k \geq s$ and $k \in \{0, 1, 2\}$
9:         Forward the model to get the predictions through Equation (6)
10:        Calculate the multiscale velocity loss $\mathcal{L}_{mv}$ as Equation (7)
11:        Run back-propagation and update network parameters
12:     **end for**
13: **end for**
14: **Return:** trained `LapFlow` model $V_\theta$

---

**Algorithm 2** `LapFlow` Sampling

---

1: **Input:** trained flow model $V_\theta$, a largest-scale random noise $\mathbf{x}_0$, critical time points $T_1$ and $T_2$
2: Get Laplacian pyramid $\{\mathbf{x}_0^{(0)}, \mathbf{x}_0^{(1)}, \mathbf{x}_0^{(2)}\}$ from $\mathbf{x}_0$
3: $\{\hat{\mathbf{x}}_{T_2}^{(2)}\} = \text{ODEINT}(V_\theta, \{0, T_2\}, \{\mathbf{x}_0^{(2)}\})$
4: $\{\hat{\mathbf{x}}_{T_1}^{(1)}, \hat{\mathbf{x}}_{T_1}^{(2)}\} = \text{ODEINT}(V_\theta, \{T_2, T_1\}, \{\sigma_{T_2}^{(1)}\mathbf{x}_0^{(1)}, \hat{\mathbf{x}}_{T_2}^{(2)}\})$
5: $\{\hat{\mathbf{x}}_1^{(0)}, \hat{\mathbf{x}}_1^{(1)}, \hat{\mathbf{x}}_1^{(2)}\} = \text{ODEINT}(V_\theta, \{T_1, 1\}, \{\sigma_{T_1}^{(0)}\mathbf{x}_0^{(0)}, \hat{\mathbf{x}}_{T_1}^{(1)}, \hat{\mathbf{x}}_{T_1}^{(2)}\})$
6: $\hat{\mathbf{x}}_1 = \hat{\mathbf{x}}_1^{(0)} + \text{Up}(\hat{\mathbf{x}}_1^{(1)}) + \text{Up}(\text{Up}(\hat{\mathbf{x}}_1^{(2)}))$
7: **Return:** the largest-scale sample $\hat{\mathbf{x}}_1$

---

three scales here for simplicity, while it can easily generalize to other numbers of scales):

$$\mathbf{x}_1^{(2)} = \text{Down}(\text{Down}(\mathbf{x}_1)), \quad \mathbf{x}_1^{(1)} = \text{Down}(\mathbf{x}_1) - \text{Up}(\mathbf{x}_1^{(2)}), \quad \mathbf{x}_1^{(0)} = \mathbf{x}_1 - \text{Up}(\text{Down}(\mathbf{x}_1)). \quad (1)$$

In this work, we use the superscript 2 to denote the smallest scale and use 0 to denote the largest scale. Given the three scales' representation, the full data $\mathbf{x}_1$ can be reconstructed from three residuals as

$$\mathbf{x}_1 = \mathbf{x}_1^{(0)} + \text{Up}(\mathbf{x}_1^{(1)}) + \text{Up}(\text{Up}(\mathbf{x}_1^{(2)})). \quad (2)$$

## 4 METHODOLOGY

### 4.1 MULTI-SCALE NOISING PROCESS

We propose learning the components of three scales at different speeds. We identify two critical time points $T_1$ and $T_2$ as two hyperparameters, where $0 \triangleq T_3 < T_2 < T_1 < 1$. In addition, we denote $T_0 = 1$ and $T_3 = 0$ for convenience of notation. For the $k$-th scale, we train the model from $t = T_{k+1}$ to $t = 1$, which means that the model of larger scales is trained for a shorter period of time. In a time step $t$ (note that $T_{k+1} \leq t \leq 1$ ensured by our progressive multi-stage training strategy, which will be presented in Section 4.2), we construct the noisy data of scale $k$ via a weighted sum of $\mathbf{x}_0^{(k)}$ and $\mathbf{x}_1^{(k)}$:

$$\mathbf{x}_t^{(k)} = \alpha_t^{(k)}\mathbf{x}_1^{(k)} + \sigma_t^{(k)}\mathbf{x}_0^{(k)}, \quad (3)$$

where $\{\alpha_t^{(k)}, \sigma_t^{(k)}\}$ are pre-defined per-scale coefficients for data and noise. The interpolation can be conducted through a linear path or a GVP path by modifying the coefficients:

$$\text{Linear:} \quad \alpha_t^{(k)} = \frac{t - T_{k+1}}{1 - T_{k+1}}, \quad \sigma_t^{(k)} = 1 - t, \quad (4a)$$

$$\text{GVP:} \quad \alpha_t^{(k)} = \sin\left(\frac{\pi(t - T_{k+1})}{2(1 - T_{k+1})}\right), \quad \sigma_t^{(k)} = \cos\left(\frac{\pi}{2}t\right). \quad (4b)$$

This multi-scale flow satisfies two key properties: (1) At the starting time $t = T_{k+1}$ for scale $k$, the noisy data contains only the scaled noise component, as $\alpha_{T_{k+1}}^{(k)} = 0$ results in $\mathbf{x}_{T_{k+1}}^{(k)} = \sigma_{T_{k+1}}^{(k)} \mathbf{x}_0^{(k)}$; (2) At the ending time $t = 1$, the representation $\mathbf{x}_1^{(k)}$ converges to the clean Laplacian residual of scale $k$, completing the generation process. The proposed flow provides the following multi-scale velocity:

$$\mathbf{u}_t^{(k)}(\mathbf{x}_t^{(k)}|\mathbf{x}_1^{(k)}) = \dot{\alpha}_t^{(k)} \mathbf{x}_1^{(k)} + \dot{\sigma}_t^{(k)} \mathbf{x}_0^{(k)}, \tag{5}$$

where $\dot{\alpha}_t^{(k)}$ and $\dot{\sigma}_t^{(k)}$ are temporal derivatives of $\alpha_t^{(k)}$ and $\sigma_t^{(k)}$ respectively.

## 4.2 Progressive Multi-stage Training and Training Targets

The training algorithm is presented in Algorithm 1. We adopt a progressive training scheme (Karras et al., 2018), which trains different-scale models for different time ranges. We introduce the concept of *stage* (denoted by $s \sim U([0, 1, 2])$, where in different stages, we train different scales of models. In training stage $s \in \{0, 1, 2\}$, we train all scales $k$ such that $k \geq s$ (noting that smaller $k$ values represent higher resolutions, with $k = 0$ being the highest resolution and $k = 2$ being the lowest). After sampling $s$, the timestep $t$ is then sampled from the range $[T_{s+1}, 1]$ (Line 6). As a result, the smallest scale ($k = 2$) is trained throughout the time range $[0, 1]$, the mid-resolution scale ($k = 1$) is trained across $[T_2, 1]$, and the highest resolution scale ($k = 0$) is trained only during $[T_1, 1]$.

Our model takes noisy multi-scale data $\{\mathbf{x}_t^{(k)}\}_{k=2}^s$ as input, and outputs corresponding multi-scale velocities $\{\mathbf{v}_t^{(k)}\}_{k=2}^s$. Both input and output consist of scales from the smallest scale ($k = 2$) to the current scale ($k = s$), while they are conditioned on the same time $t$. Let $V_\theta$ be the `LapFlow` model (see Section 4.4 for a detailed architecture), the model prediction (Line 9) is formulated as:

$$\{\mathbf{v}_t^{(k)}\}_{k=2}^s = V_\theta \left( \{\mathbf{x}_t^{(k)}\}_{k=2}^s \right). \tag{6}$$

We apply a multiscale conditioning flow matching loss (Line 10) as:

$$\mathcal{L}_{mv} = \sum_{k=2}^s w_k \mathbb{E}_{t, q(\mathbf{x}_1^{(k)}), p_t(\mathbf{x}_t^{(k)}|\mathbf{x}_1^{(k)})} \|\mathbf{v}_t^{(k)} - \mathbf{u}_t^{(k)}(\mathbf{x}_t^{(k)}|\mathbf{x}_1^{(k)})\|^2, \tag{7}$$

where $w_k$ is a balancing factor for scale $k$'s loss, where in practice we set $w_k = 1$ for simplicity.

## 4.3 Multi-scale Sampling Process

We present our sample algorithm in Algorithm 2, which is used to generate images with the highest resolution. The sampling algorithm starts with a Laplacian of noise (Line 2). We then leverage the ODE solving function ODEINT, which takes three arguments (a model $V_\theta$, initial and end timesteps $\{t_i, t_e\}$, and an initial condition $\mathbf{x}_{t_i}$) as input and outputs the variable $\mathbf{x}_{t_e}$ corresponding to the end timestep $t_e$. A popular implementation of ODEINT that we use is the torchdiffeq library (Chen et al., 2018). This general process can be formally expressed as $\mathbf{x}_{t_e} = \text{ODEINT}(V_\theta, \{t_i, t_e\}, \mathbf{x}_{t_i})$.

We first denoise the smallest scale from $t = 0$ to $T_2$ to get the partially denoised state $\mathbf{x}_{T_2}^{(2)}$ (Line 3). Then we continue the denoising procedure during $T_2 \leq t < T_1$, where we denoise the mid-scale and smallest scale in parallel (Line 4). In this step, we leverage the fact that when $t = T_2$, the mid-resolution noisy state becomes $\sigma_{T_2}^{(1)} \mathbf{x}_0^{(1)}$, while the smallest noisy state becomes $\hat{\mathbf{x}}_{T_2}^{(2)}$ according to Equation (3). Therefore, this step would take an initial condition $\{\sigma_{T_2}^{(1)} \mathbf{x}_0^{(1)}, \hat{\mathbf{x}}_{T_2}^{(2)}\}$. Similarly, the last step would take $\{\sigma_{T_1}^{(1)} \mathbf{x}_0^{(0)}, \hat{\mathbf{x}}_{T_1}^{(1)}, \hat{\mathbf{x}}_{T_1}^{(2)}\}$ as the initial condition. We finally evaluate the model from $T_1$ to 1 (Line 5), and these denoised residuals are then combined into full-scale data via Equation (2).

## 4.4 Multi-scale DiT with Mixture-of-Transformers

We present the `LapFlow` model, which is a multi-scale diffusion transformer (DiT) with Mixture-of-Transformers (MoT (Liang et al., 2024)). The complete model architecture of `LapFlow` is shown in Figure 2. As formulated in Equation (6), our `LapFlow` architecture is designed with flexible input-output capabilities: it can process noisy states of any subset of scales (from one to three scales) as input, and will correspondingly output velocity predictions for exactly the same

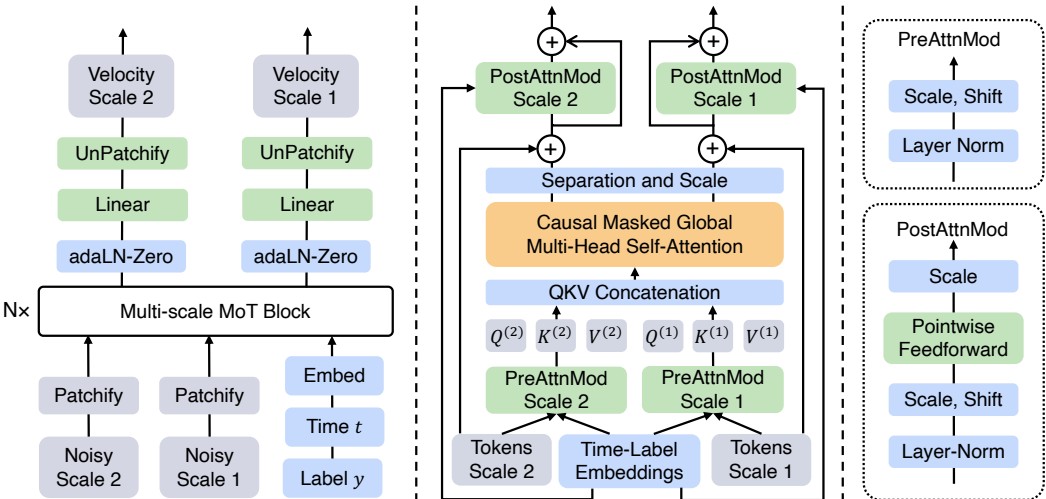

Figure 2: (**Left:**) Schematic of the `LapFlow` model $V_\theta$. The multi-scale transformer takes multi-scale noisy states as input, conditioned on time and label, and predicts velocities for each input scale. While the model can take an arbitrary number of scales as input, we show two here for simplicity. (**Middle:**) Details of one multi-scale MoT block. We use separate QKVs for different scales, while the attention is computed globally. Furthermore, we adopt a mask to enforce causal relationships across scales. (**Right:**) Details of scale-specific `PreAttnMod` and `PostAttnMod` modules (Peebles and Xie, 2023), where each `PostAttnMod` module includes a feedforward network (FFN).

scales that were provided as input. This flexibility enables the model to handle different stages of the generation process where varying numbers of scales are active. We inherit best practices from DiT (Peebles and Xie, 2023), including patchify, feature modulations, and in-context conditioning. In addition, we utilize MoT (Liang et al., 2024) to enable scale-specific processing (e.g., pre-attention and post-attention modulations) with global multi-head self-attention over all scales. In the global attention mechanism, `LapFlow` implements a causal masking strategy that enforces a unidirectional information flow from smaller scales (lower resolutions) to larger scales (higher resolutions).

**Model schematic (left of Figure 2).** The input noisy states at each scale are first converted into sequences of tokens using the `Patchify` operation (Peebles and Xie, 2023). For each scale, sine-cosine positional embeddings (Dosovitskiy et al., 2020) are added to its corresponding tokens. Conditions such as time $t$ and label $y$ are added as additional tokens to the input sequence (in-context conditioning (Peebles and Xie, 2023), enabling flexible conditioning without architectural modifications. The input tokens are then processed by $N$ multi-scale MoT blocks, followed by an adaptive Layer Norm layer with zero initialization (`adaLN-Zero` (Liang et al., 2024)). Finally, a linear decoder and an `UnPatchify` (rearrange) process are applied to get velocity predictions.

**Multi-scale MoT block (middle and right of Figure 2).** For any scale $k \in \{s, ..., 2\}$ (where $s$ is the current training stage), we use $\alpha_1^{(k)}, \alpha_2^{(k)}, \beta_1^{(k)}, \beta_2^{(k)}, \gamma_1^{(k)}$ and $\gamma_2^{(k)}$ to denote parameters for scales and shifts regressed from conditional embeddings, following DiT convention (Peebles and Xie, 2023). Firstly, we apply a pre-attention modulation process (`PreAttnMod`) that includes both scale-and-shift operations through factors $\gamma_1^{(k)}$ and $\beta_1^{(k)}$, as well as scale-specific projections to obtain queries, keys, and values (QKVs). Let $z^{(k)}$ be the tokens after the scale-and-shift, and the scale-specific QKVs are:

$$Q^{(k)} = z^{(k)} W_Q^{(k)}, K^{(k)} = z^{(k)} W_K^{(k)}, V^{(k)} = z^{(k)} W_V^{(k)}, \tag{8}$$

where $W_Q^{(k)}, W_K^{(k)}$ and $W_V^{(k)}$ are scale-specific weight matrices. Then, a masked global multi-head self-attention is applied to concatenations of QKVs from all scales. The single-head formulation is:

$$\texttt{MaskedGlobalAttn}(Q, K, V) = \texttt{Softmax}\left(\frac{QK^\top}{\sqrt{d}} + M_c\right) V, \tag{9}$$

Table 1: **Evaluations on CelebA-HQ.** For all multi-scale methods (EdifyImage (Atzmon et al., 2024), Pyramidal Flow (Jin et al., 2025), and ours), we report averaged GFLOPs and time during inference.

| Methods | Backbone | Resolution | Space | FID↓ | Recall↑ | NFE | Time (s) | Params | GFLOPs |
|---|---|---|---|---|---|---|---|---|---|
| *Standard Resolution (256)* | | | | | | | | | |
| FM (Lipman et al., 2023) | U-Net | $256 \times 256$ | Image | 7.34 | - | 128 | - | - | - |
| LDM (Rombach et al., 2022) | U-Net | $256 \times 256$ | Latent | 5.11 | 0.49 | 50 | 2.90 | 150M | 10.2 |
| LFM (Dao et al., 2023) | DiT-L/2 | $256 \times 256$ | Latent | 5.26 | 0.46 | 89 | 1.70 | 541M | 22.1 |
| Pyramidal Flow (Jin et al., 2025) | DiT-L/2 | $256 \times 256$ | Latent | 11.20 | 0.48 | 90 | 1.85 | 544M | 14.2 |
| EdifyImage (Atzmon et al., 2024) | DiT-L/2 | $256 \times 256$ | Image | 7.62 | 0.47 | 95 | 2.10 | 602M | 28.9 |
| Ours | DiT-L/2 | $256 \times 256$ | Latent | 3.53 | 0.53 | 80 | 1.51 | 612M | 16.5 |
| *High Resolution (512 and 1024)* | | | | | | | | | |
| LFM (Dao et al., 2023) | DiT-L/2 | $512 \times 512$ | Latent | 6.35 | 0.41 | 93 | 2.90 | 544M | 43.5 |
| Ours | DiT-L/2 | $512 \times 512$ | Latent | 4.04 | 0.47 | 85 | 2.60 | 680M | 41.7 |
| LFM (Dao et al., 2023) | DiT-L/2 | $1024 \times 1024$ | Latent | 8.12 | 0.34 | 100 | 4.20 | 544M | 154.8 |
| Ours | DiT-L/2 | $1024 \times 1024$ | Latent | 5.51 | 0.39 | 94 | 3.30 | 680M | 148.2 |

where $Q = [Q^{(s)}; \ldots; Q^{(2)}]$, $K = [K^{(s)}; \ldots; K^{(2)}]$, $V = [V^{(s)}; \ldots; V^{(2)}]$ are concatenations of scale-specific QKVs, and $d$ is the feature dimension. We use a block causal mask (Tian et al., 2024) $M_c$ to ensure that each scale $k$ can only attend to scales smaller than or equal to $k$ ($k' \geq k$). The output of the global attention, a unified sequence of token representations, is subsequently partitioned according to scale, yielding separate feature sequences for each scale. Each scale's feature is then scaled by $\alpha_1^{(k)}$. Finally, we apply the `PostAttnMod` module, which consists of layer normalization, a scale-and-shift operation (using $\gamma_2^{(k)}$ and $\beta_2^{(k)}$), a point-wise feed-forward network (FFN), and a final scaling by $\alpha_2^{(k)}$. Residual connections are applied to connect scale-wise representations.

## 5 EXPERIMENTS

### 5.1 EXPERIMENTAL SETUP

**Datasets.** We conduct experiments on two datasets: CelebA-HQ (Karras et al., 2018) and ImageNet (Deng et al., 2009). We use the term "resolution" to indicate the largest-scale resolution we are trying to generate (which is the resolution of the output to our sampling algorithm Algorithm 2). For CelebA-HQ, we experiment with resolutions $256 \times 256$, $512 \times 512$, and $1024 \times 1024$. For ImageNet, the resolution is set at $256 \times 256$ due to limited resources. CelebA-HQ is unconditional, while ImageNet is conditioned on class labels. The VAE down-sample factor is eight, so the largest latent is $32 \times 32$. We experiment with varied classifier-free guidance (CFG) (Ho and Salimans, 2022) on ImageNet, while CFG is not enabled by default.

**Baselines.** We benchmark and compare against several flow matching baselines on CelebA-HQ, including LFM (Dao et al., 2023), which is a single-scale baseline. We also compare with a recent multi-scale method called `PyramidalFlow` (Jin et al., 2025) in the task of image generation. Besides, we compare with `EdifyImage`, which is an image-space method using multi-scale generation. We re-implement `EdifyImage` and `PyramidalFlow` under the flow matching framework for comparison. On ImageNet, we further evaluate our model against a broad range of recent works in both diffusion and flow matching paradigms. All models use DiT-L/2 for CelebA-HQ, and we experiment with DiT-B/2 and DiT-XL/2 for ImageNet. We utilize the Dormand-Prince method (dopri5) implemented in the torchdiffeq library (Chen et al., 2018). We adopt the Fréchet inception distance (Heusel et al., 2017) with $50K$ images as the main metric.

**VAE.** While a popular choice of VAE is the SDVAE (Rombach et al., 2022), a recent work called EQVAE (Kouzelis et al., 2025) introduces an equivariant regularization technique that enhances latent space structure by preserving scaling transformations. In practice, we find that EQVAE greatly benefits our multi-scale approach because it offers equivalent representations across scales. Since EQVAE is only trained on $256 \times 256$, we use SDVAE instead for higher resolutions.

**Hyperparameters and training settings.** For experiments that generate resolutions $256 \times 256$, we use **two levels** in all multi-scale methods (`PyramidalFlow`, `EdifyImage`, and ours), where

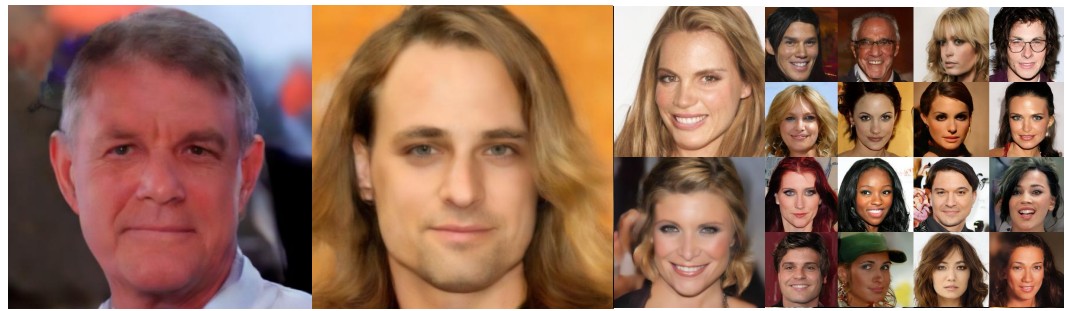

Figure 3: Qualitative results on CelebA-HQ 1024 (left two), 512 (middle two), and 256 (right).

Table 2: **Ablation studies on CelebA-HQ.** We analyze various design choices, including VAE architecture, mixture-of-Transformer (MoT) design, noise schedule, and other modeling configurations. We report FID-50K scores on $256 \times 256$ resolution images. Our default settings are marked in  gray .

(a) **VAE choices.** EQVAE (Kouzelis et al., 2025) benefits our method over SDVAE (Rombach et al., 2022) but it cannot improve LFM (Dao et al., 2023).

| Method | LFM (SDVAE) | LFM (EQVAE) | Ours (SDVAE) | Ours (EQVAE) |
|---|---|---|---|---|
| FID | 5.26 | 7.77 | 4.37 | **3.53** |

(b) **MoT design.** Comparison between separate weights models and our proposed MoT model, the MoT can reduce the GFLOPs by selecting experts.

| Metric | Separate Model | MoT |
|---|---|---|
| GFLOPs | 38.9 | **16.5** |
| FID | 3.60 | **3.53** |

(c) **Causal masking.** "None" means no mask is used, "Self" means each scale is only attended on itself.

| Mask | None | Self | Causal |
|---|---|---|---|
| FID | 3.91 | 5.19 | **3.53** |

(d) **Critical time point $T$.** Compare $T$ values, which determines the temporal segments of two scales.

| $T$ | 0.1 | 0.2 | 0.5 | 0.9 |
|---|---|---|---|---|
| FID | 5.12 | 4.37 | **3.53** | 4.92 |

(e) **Learning rate decay final LR.** The final learning rate of $1 \times 10^{-6}$ achieves optimal performance, with both higher and lower values degrading results.

| Final LR | $2 \times 10^{-4}$ | $1 \times 10^{-5}$ | $1 \times 10^{-6}$ | $1 \times 10^{-7}$ |
|---|---|---|---|---|
| FID | 4.12 | 3.81 | **3.53** | 4.82 |

(f) **Noise schedule.** The simple linear schedule outperforms the GVP in our model, refer to Equation (4).

| Noise Schedule | LFM (Linear) | LFM (GVP) | Ours (Linear) | Ours (GVP) |
|---|---|---|---|---|
| FID | 5.26 | 4.39 | **3.53** | 4.10 |

(g) **Number of scales.** Two-scale representation achieves optimal performance in $256 \times 256$, with more scales increasing complexity without further gains.

| # of Scales | 1 (LFM) | 2 (Ours) | 3 (Ours) | 4 (Ours) |
|---|---|---|---|---|
| FID | 5.26 | **3.53** | 3.59 | 5.12 |

(h) **Modeling Space.** Latent space modeling significantly outperforms image space for both approaches, with our method showing substantial gains in both.

| Method | LFM (Image) | LFM (Latent) | Ours (Image) | Ours (Latent) |
|---|---|---|---|---|
| FID | 11.58 | 5.26 | 8.63 | **3.53** |

the default time segment is $T = 0.5$. We use three levels for cases where resolutions are higher than $256 \times 256$, where time segments hyperparameters are $T_2 = 0.33$ and $T_1 = 0.67$. We use the linear scheduler instead of GVP unless explicitly mentioned. For ImageNet $256 \times 256$, we use a constant lr of $1 \times 10^{-4}$, a global batch size of 256, and an EMA decay of 0.9999. We train $600K$ steps for both the B/2 model and the XL/2 model. We further train the B/2 model until $7M$ steps following LFM (Dao et al., 2023). For CelebA-HQ, we adopt a CosineAnnealingLR (Loshchilov and Hutter, 2017) scheduler with an initial learning rate of $2 \times 10^{-4}$ and a final learning rate of $1 \times 10^{-6}$. Training durations vary across resolutions, as higher resolutions typically require more iterations to converge. A complete list of training hyperparameters can be found in Appendix G. All models are trained on a computing node with 8 NVIDIA H200 GPUs.

## 5.2 EXPERIMENTAL RESULTS

**Unconditional generation on CelebA-HQ.** We evaluate our method on the CelebA-HQ dataset at resolutions of $256 \times 256$, $512 \times 512$, and $1024 \times 1024$, comparing against single-scale and multi-scale flow matching approaches. As shown in Table 1, our method achieves an FID of 3.53 at $256 \times 256$,

Table 3: **Results of class-conditional generation on ImageNet** $256 \times 256$.

| Methods | Backbone | Training Steps | CFG | FID | NFE | Time (s) | GFLOPs |
|---|---|---|---|---|---|---|---|
| DiT (Peebles and Xie, 2023) | DiT-B/2 | 600K | $\times$ | 41.27 | 250 | 1.87 | 5.6 |
| LFM (Dao et al., 2023) | DiT-B/2 | 600K | $\times$ | 48.29 | 89 | 1.70 | 5.6 |
| Pyramidal Flow (Jin et al., 2025) | DiT-B/2 | 600K | $\times$ | 39.40 | 90 | 1.74 | 5.7 |
| Ours | DiT-B/2 | 600K | $\times$ | **36.50** | 80 | 1.25 | 4.9 |
| DiT (Peebles and Xie, 2023) | DiT-XL/2 | 600K | $\times$ | 19.50 | 250 | 3.50 | 29.1 |
| LFM (Dao et al., 2023) | DiT-XL/2 | 600K | $\times$ | 28.37 | 89 | 3.32 | 29.1 |
| Pyramidal Flow (Jin et al., 2025) | DiT-XL/2 | 600K | $\times$ | 17.10 | 90 | 4.02 | 29.9 |
| Ours | DiT-XL/2 | 600K | $\times$ | **14.38** | 80 | 2.85 | 20.5 |
| LFM (Dao et al., 2023) | DiT-B/2 | 7M | $\times$ | 20.38 | 89 | 1.70 | 5.6 |
| LFM (Dao et al., 2023) | DiT-B/2 | 7M | 1.5 | 4.46 | 89 | 1.70 | 5.6 |
| Ours | DiT-B/2 | 7M | $\times$ | 18.55 | 80 | 1.25 | 4.9 |
| Ours | DiT-B/2 | 7M | 1.5 | **4.12** | 80 | 1.25 | 4.9 |

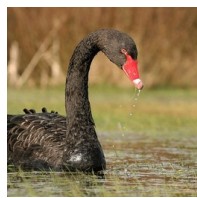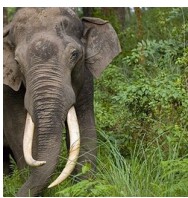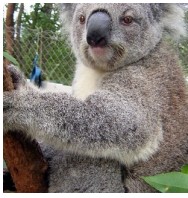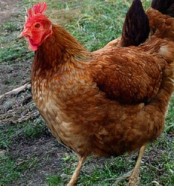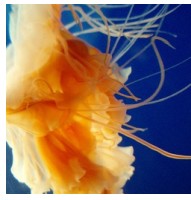

Figure 4: Qualitative results on ImageNet $256 \times 256$ using our trained B/2 model with CFG=1.5.

outperforming LFM (5.26) and Pyramidal Flow (11.20). The performance gap becomes even more pronounced at higher resolutions, where our method achieves FID scores of 4.04 and 5.51 at $512 \times 512$ and $1024 \times 1024$, respectively, compared to 6.35 and 8.12 for LFM. In particular, our method not only produces higher quality samples, but also requires fewer function evaluations (NFE) and less inference time compared to flow matching baselines (`LFM` (Dao et al., 2023), `PyramidalFlow` (Jin et al., 2025), and `EdifyImage` (Atzmon et al., 2024)), demonstrating the efficiency of our multi-scale architecture in preserving image fidelity while reducing computational requirements. While Relay Diffusion obtains a competitive FID of 3.15 at $256 \times 256$, it relies on image-space modeling and a U-Net backbone, leading to significantly higher inference cost (1221 GFLOPs vs. 16.5 GFLOPs for ours). This indicates a fundamentally different computational regime, as our latent-space DiT architecture provides a far more favorable tradeoff between image fidelity and efficiency. Furthermore, Relay Diffusion is strictly formulated as a two-stage framework and, to our knowledge, has not been demonstrated beyond $256 \times 256$ resolutions, whereas our Laplacian multi-scale formulation scales effectively to $512 \times 512$ and $1024 \times 1024$ while retaining efficiency gains. Figure 3 shows qualitative results of our method across different resolutions, highlighting high-fidelity detail preservation and demonstrating that our multi-scale formulation is effective even at megapixel resolution.

**Ablation studies.** Our comprehensive ablation studies on CelebA-HQ ($256 \times 256$ resolution) demonstrate several key insights for `LapFlow`. As shown in Table 2a, the choice of VAE architecture significantly impacts performance, with EQVAE providing substantial benefits to our approach while proving detrimental to the baseline LFM. The proposed mixture-of-transformers (MoT) design (Table 2b) achieves both computational efficiency and improved generation quality compared to separate models. Furthermore, we find causal masking is optimal among attention strategies (Table 2c). We then ablate the choice of critical time points. For two-scale experiments, we observe a critical time point of $T = 0.5$ (Table 2d) strikes the ideal balance between model expressivity and complexity. Here we want to note that in two scales there is only one critical time point $T$. As summarized in Table 2f and further discussed in Appendix D, the linear pair $\sigma_t^{(k)} = 1 - t$ and $\alpha_t^{(k)} = \frac{t - T_{k+1}}{1 - T_{k+1}}$ achieves the strongest performance, outperforming quadratic and cubic decay as well as the GVP formulation. As shown in Table 2g, two scales yield the best performance at 256Œ256. This is due to the latent resolution being only $32 \times 32$; adding a third scale would require an $8 \times 8$ stage, which we find too small to provide reliable semantic guidance during early flow integration. To further validate this resolution-dependent behavior, we extend this ablation to $512^2$ and $1024^2$ generation. As

shown in Appendix B, larger latent grids (64×64 and 128×128) benefit from a more-scale hierarchy, confirming that higher resolutions may require more number of scales of `LapFlow`. A similar trend is reported in the concurrent work PixelFlow (Chen et al., 2025), which observes degraded performance when the initial kickoff sequence becomes too short, although their model operates in pixel space rather than latent space. To validate temporal segmentation beyond the two-scale case, we additionally ablate three-scale configurations and show that a balanced time allocation across scales yields the best results; see Appendix A for FID comparisons at $1024 \times 1024$. We further study the effect of the noise and coupling schedules used in our flow parameterization. Specifically, we compare the linear transport and noise decays with alternative polynomial and GVP-based schedules. These results support our choice of simple scale-wise linear schedules for stable multi-scale flow evolution. The performance advantage of latent space modeling instead of the image space counterpart is particularly noteworthy (Table 2h), where our approach substantially outperforms both image-space implementations and the LFM baseline in two domains. These findings, combined with our optimized learning rate decay with a final LR of $1 \times 10^{-6}$ (Table 2e), collectively contribute to our full model.

**Class-conditional generation on ImageNet.** As illustrated in Figure 5, our method consistently shows superior performance compared to LFM (Dao et al., 2023) for both B/2 and XL/2 models. We present numerical results in Table 3 and qualitative results in Figure 4. Our approach achieves the best FID score of 36.50 with the B/2 backbone and 14.38 with the XL/2 backbone at 600K training steps, outperforming all baseline methods including DiT (Peebles and Xie, 2023), LFM (Dao et al., 2023), and Pyramidal Flow (Jin et al., 2025). Notably, our method works under fewer NFE (80 vs. 89-250) and demonstrates improved computational efficiency with lower inference time (1.25s vs. 1.70-1.87s for B/2) and reduced GFLOPs (4.9 vs. 5.6-5.7 for B/2). When trained for 7M steps with classifier-free guidance (CFG=1.5), our approach achieves an FID score of 4.12, surpassing LFM's 4.46 while being more efficient.

## 6 CONCLUSION

In this paper, we presented `LapFlow`, a novel framework that advances generative modeling by integrating parallel multi-scale generation within the flow matching paradigm. Our approach decomposes images into Laplacian pyramid residuals and processes them simultaneously through a specialized MOT architecture with causal attention mechanisms that enforce hierarchical information flow. Extensive experimental results on CelebA-HQ and ImageNet datasets demonstrate that our method consistently achieves superior FID scores than prior flow matching works while requiring fewer function evaluations and reduced computational resources during sampling. Performance advantages are particularly pronounced at higher resolu-

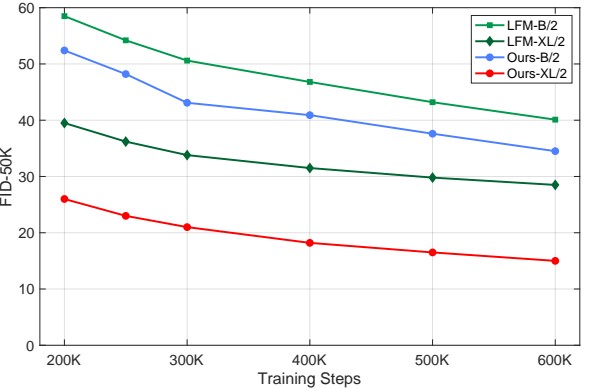

Figure 5: FID-50K on ImageNet (256×256) across training iterations comparing LFM (Dao et al., 2023) with ours using two backbones (B/2 and XL/2).

tions (up to 1024Œ1024), demonstrating its superior scalability for complex, detail-rich generation tasks even with limited training resources. An interesting future direction is to integrate advanced training accelerators such as REPA (Yu et al., 2025), which has demonstrated significantly faster convergence for flow-based generative models. We expect that combining training accelerators with the multi-scale Laplacian formulation could further reduce training cost.

## 7 ACKNOWLEDGMENTS

The authors are supported in part by grant NSF 2409016.

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

APPENDIX

## A  TEMPORAL SEGMENTATION THRESHOLD VALIDATION

As discussed in the main paper, our progressive multi-scale diffusion training strategy allocates timesteps across spatial scales using temporal segmentation thresholds. While Table 2(d) presents an ablation on a single segmentation point $T$ in a two-scale setting, we provide additional justification for the selected three-scale thresholds. To address this concern, we provide extended ablation experiments on the temporal segmentation thresholds $T_1$ and $T_2$ for a three-scale configuration at a resolution of $1024 \times 1024$. Except for the threshold values, all other training configurations are identical to Table 2.

Results are presented in Table 5. These results empirically validate our threshold choice in the main paper, demonstrating that a balanced temporal allocation across scales ($T_1 = 0.67, T_2 = 0.33$) yields the best generation quality. This trend is consistent with our two-scale observations, where overly early or late transitions led to degraded FID.

We hypothesize that allocating sufficient diffusion steps to the coarse-scale stage stabilizes global structures before introducing high-frequency refinements in later stages. In contrast, excessively delayed transitions (e.g., $T_1 = 0.90$) limit the optimization of high-resolution layers, while overly early transitions (e.g., $T_2 = 0.10$) destabilize global consistency and lead to artifacts.

A complete grid search for optimal segmentation is computationally prohibitive at high resolutions; however, the reported results provide strong empirical justification for our selected thresholds.

**Practical Recommendation.** Based on this analysis, we recommend the following default heuristic for future multi-scale diffusion training:

$$T_1 \approx \frac{2}{3}, \qquad T_2 \approx \frac{1}{3}, \tag{10}$$

which we found to consistently improve stability and generation quality.

## B  ADDITIONAL ANALYSIS: NUMBER OF SCALES AT HIGHER RESOLUTIONS

In the main paper, we show that at $256^2$ resolution, a two-scale Laplacian hierarchy achieves the best performance. Here, we extend this analysis to higher resolutions.

As the resolution increases, the VAE latent grids become larger:

$$256^2 \to 32 \times 32, \quad 512^2 \to 64 \times 64, \quad 1024^2 \to 128 \times 128.$$

We present a further study at larger scales. Other unmentioned setups are the same as Table 2g.

Table 4: **FID vs. number of Laplacian scales** across 2561024 resolutions on CelebA-HQ.

| Resolution | Latent size | # Scales | Decomposition | FID ↓ |
|---|---|---|---|---|
| 256×256 | 32×32 | 2 | $32 \to 16$ | **3.53** |
| 256×256 | 32×32 | 3 | $32 \to 16 \to 8$ | 3.59 |
| 256×256 | 32×32 | 4 | $32 \to 16 \to 8 \to 4$ | 5.12 |
| 512×512 | 64×64 | 2 | $64 \to 32$ | 5.45 |
| 512×512 | 64×64 | 3 | $64 \to 32 \to 16$ | **4.04** |
| 512×512 | 64×64 | 4 | $64 \to 32 \to 16 \to 8$ | 4.12 |
| 1024×1024 | 128×128 | 2 | $128 \to 64$ | 6.62 |
| 1024×1024 | 128×128 | 3 | $128 \to 64 \to 32$ | 5.51 |
| 1024×1024 | 128×128 | 4 | $128 \to 64 \to 32 \to 16$ | **5.45** |

At low resolution, additional Laplacian levels produce coarse grids too small to stabilize flow trajectories, whereas at higher resolution, three-scale Laplacian modeling delivers consistent improvements. These results are consistent with observations in concurrent PixelFlow (Chen et al., 2025), which reports degraded performance when early-stage spatial support becomes too small, despite operating in pixel (not latent) space.

| Resolution | $T_1$ | $T_2$ | FID↓ |
|---|---|---|---|
| 1024×1024 | 0.67 | 0.33 | **5.51** |
| 1024×1024 | 0.90 | 0.33 | 6.72 |
| 1024×1024 | 0.67 | 0.10 | 8.12 |

Table 5: Three-scale temporal segmentation ablation at $1024^2$ resolution. Balanced allocation ($T_1 = 0.67, T_2 = 0.33$) achieves the best FID, confirming the trend observed in the two-scale case.

## C  TIME-WEIGHTED COMPLEXITY ANALYSIS OF MULTI-SCALE MoT

We provide a formal complexity analysis of the proposed Mixture-of-Transformers (MoT) architecture, taking into account that different Laplacian scales are active over different segments of the ODE time domain. Although global attention is applied over the concatenated token sequence of all active scales at each time segment, the progressively shortened active ranges of higher-resolution scales reduce the time-averaged number of tokens processed during sampling.

Under the standard quadratic attention cost used in transformer analysis (Vaswani et al., 2017), the per-layer cost scales as $N^2 d$, where $N$ is the number of latent tokens and $d$ is the embedding dimension; full self-attention scales quadratically with sequence length.

Let the finest latent resolution be $H \times W$ with downsampling factor $s$, so the largest-scale latent has $N = \frac{HW}{s^2}$ tokens. Following the Laplacian decomposition in Equation (2), the *per-scale* token counts at the three spatial resolutions are $N^{(0)} = N$, $N^{(1)} = N/4$, and $N^{(2)} = N/16$. During sampling, the ODE is solved in three segments: from $t \in [0, T_2)$ only the coarsest scale is active, from $t \in [T_2, T_1)$ the coarsest and mid scales are active, and from $t \in [T_1, 1]$ all three scales are active. Thus, the total number of tokens processed in each segment is

$$N_2 = N^{(2)} = \tfrac{N}{16}, \qquad N_1 = N^{(1)} + N^{(2)} = \tfrac{5N}{16}, \qquad N_0 = N^{(0)} + N^{(1)} + N^{(2)} = \tfrac{21N}{16}. \quad (11)$$

Let the segment lengths be $\Delta t_2 = T_2 - 0$, $\Delta t_1 = T_1 - T_2$, and $\Delta t_0 = 1 - T_1$. With our default choice $T_2 \approx \tfrac{1}{3}$ and $T_1 \approx \tfrac{2}{3}$ (Sec. 4.3), we have $\Delta t_2 \approx \Delta t_1 \approx \Delta t_0 \approx \tfrac{1}{3}$. Because attention cost in the $k$-th segment is proportional to $N_k^2 d$, the time-weighted attention complexity of one MoT layer over the whole ODE trajectory is

$$\text{Cost}_{\text{MoT}} \;\propto\; \Delta t_2 N_2^2 d + \Delta t_1 N_1^2 d + \Delta t_0 N_0^2 d \;\approx\; \tfrac{1}{3}\left[\left(\tfrac{N}{16}\right)^2 + \left(\tfrac{5N}{16}\right)^2 + \left(\tfrac{21N}{16}\right)^2\right] d \approx 0.61\, N^2 d.$$
$$(12)$$

Therefore, under our default three-scale configuration the MoT layer enjoys a reduction factor of about $0.61$ in its attention term, i.e., roughly a $1.6\times$ lower attention cost than a single-scale DiT with the same finest latent size.

We emphasize that this is an idealized, time-weighted analysis focusing on the dominant quadratic attention term. Empirical GFLOPs and runtime may differ from this constant factor due to (i) the adaptive ODE solver (DormandPrince) producing different numbers of function evaluations across samples, (ii) kernel-level hardware efficiency, (iii) upsampling and residual fusion costs in Laplacian reconstruction, and (iv) framework-dependent attention kernel implementations. Nevertheless, the observed improvements in Tables 1, 2, and 3 are consistent with this theoretical reduction in effective attention cost.

## D  ADDITIONAL DISCUSSION ON LINEAR TRANSPORT AND NOISE SCHEDULES

**Clarification of schedule formulation.** Both the noise decay schedule $\sigma_t^{(k)} = 1 - t$ and the transport (coupling) schedule $\alpha_t^{(k)} = \frac{t - T_{k+1}}{1 - T_{k+1}}$ are **linear functions of** $t$. In particular,

$$\alpha_t^{(k)} = \frac{t - T_{k+1}}{1 - T_{k+1}} = \frac{1}{1 - T_{k+1}}\, t - \frac{T_{k+1}}{1 - T_{k+1}}, \qquad (13)$$

which is a linear mapping that increases from 0 to 1 as $t$ moves from $T_{k+1}$ to 1. The two linear schedules play distinct roles: $\sigma_t^{(k)}$ is globally shared across all scales, while $\alpha_t^{(k)}$ provides a shift–rescale transformation that activates Laplacian levels gradually after their transition threshold $T_{k+1}$. This design offers a simple and stable mechanism for connecting consecutive Laplacian scales *without* requiring re-noising steps used in related multi-scale flow models such as Jin et al. (2025).

**Ablation on alternative schedule choices.** To support this design choice, we further evaluate several alternative combinations of $\sigma_t^{(k)}$ and $\alpha_t^{(k)}$ at the $256 \times 256$ resolution. This experiment extends the analysis in Table 2f while keeping all other settings identical to the default configuration in Table 2.

| Noise schedule $\sigma_t^{(k)}$ | Coupling schedule $\alpha_t^{(k)}$ | FID $\downarrow$ ($256^2$) |
|---|---|---|
| $1 - t$ (Linear) | $\frac{t-T_{k+1}}{1-T_{k+1}}$ (Linear) | **3.53** |
| $\cos\left(\frac{\pi}{2}t\right)$ (GVP) | $\sin\left(\frac{\pi(t-T_{k+1})}{2(1-T_{k+1})}\right)$ (GVP) | 4.10 |
| $(1 - t)^2$ (Quadratic) | $\frac{t-T_{k+1}}{1-T_{k+1}}$ (Linear) | 3.78 |
| $(1 - t)^3$ (Cubic) | $\frac{t-T_{k+1}}{1-T_{k+1}}$ (Linear) | 4.25 |

These results indicate that the simple linear schedule $\sigma_t^{(k)} = 1 - t$ achieves the best overall performance, while steeper decays such as $(1 - t)^2$ and $(1 - t)^3$ lead to noticeably worse sample quality. The GVP variant also performs competitively, but the linear pair $(\sigma_t^{(k)}, \alpha_t^{(k)})$ is both *simpler* and *more stable* across a wide range of settings, reinforcing the choice of linear schedules as the default configuration in our framework.

# E  EDIFYIMAGE FORMULATION DETAILS

The noising process is described as a weighted interpolation between the clean image $\mathbf{x}_1$ and the standard Gaussian noise $\mathbf{x}_0^{(0)} \sim \mathcal{N}(0, I^{(0)})$ (note this is not a residual) corresponding to the largest scale, where we get a noisy state $\mathbf{x}_t^{(k)}$ for scale $k = 0, 1, 2$ via:

$$\mathbf{x}_t^{(k)} = (1 - t)\text{Down}(\mathbf{x}_0^{(0)}, 2^k) + \mu^{(k)}(\mathbf{x}_1, t), \tag{14}$$

where we use a factor $2^k$ for better compatibility of sampling across multiple scales, and we provide more discussions on this in the supplementary.

Based on the noisy target $\mathbf{x}_t^{(k)}$, we construct the following ground-true velocity:

$$\begin{aligned}
\mathbf{u}_t^{(k)}(\mathbf{x}_t^{(k)}|\mathbf{x}_1) &\triangleq \frac{d\mathbf{x}_t^{(k)}}{dt} \\
&= -\text{Down}(\mathbf{x}_0^{(0)}, 2^k) + \frac{d\mu^{(k)}(\mathbf{x}_1, t)}{dt},
\end{aligned} \tag{15}$$

where the last term is crucial because the data mean $\mu^{(k)}(\mathbf{x}_1, t)$ changes through timestep $t$. Based on this velocity, the model is trained through the following flow-matching target (Lipman et al., 2023):

$$\mathbb{E}_{t\in[T_{k+1},1],q(\mathbf{x}_1),p_t(\mathbf{x}_t^{(k)}|\mathbf{x}_1)} \left\| \mathbf{v}_t^{(k)}(\mathbf{x}_t^{(k)}) - \mathbf{u}_t^{(k)}(\mathbf{x}_t^{(k)}|\mathbf{x}_1) \right\|^2, \tag{16}$$

where $t$ is sampled from $[T_{k+1}, 1]$ (we define $T_3 \triangleq 0$, and $T_0 \triangleq 1$), $p_t(\mathbf{x}_t^{(k)}|\mathbf{x}_1)$ is the underlying conditional probability path towards $\mathbf{x}_1$ determined by $\mathbf{u}_t^{(k)}(\mathbf{x}_t^{(k)}|\mathbf{x}_1)$, and $\mathbf{v}_t^{(k)}(\mathbf{x}_t^{(k)})$ is the multi-scale machine learning model to model the flow.

# F  PYRAMIDALFLOW

---

**Algorithm 3** Pyramidal Flow Training

---

1: **Input:** Dataset of images $D$, number of stages $K$, number of epochs $M$, list of $K$ start and end times $s_k, e_k$
2: **Output:** Trained flow model $\mathbf{v}(\mathbf{x}, t)$.
3: Initialize the model weights randomly
4: **for** epoch= 1 to $M$ **do**
5:     **for** clean data $\mathbf{x}_1$ in $D$ **do**
6:         Determine the stage k for $\mathbf{x}_1$
7:         Sample a timestep $t \sim U(s_k, e_k)$
8:         Sample the start point $\mathbf{x}_0 \sim \mathcal{N}(0, I)$
9:         Down-sample noise $\mathbf{x}_0^{(k)} = \text{Down}(\mathbf{x}_0, 2^k)$
10:        Compute start points as $\mathbf{x}_s^{(k)} = s_k \cdot \text{Up}(\text{Down}(\mathbf{x}_1, 2^{k+1})) + (1 - s_k) \cdot \mathbf{x}_0^{(k)}$
11:        Compute endpoints as $\mathbf{x}_e^{(k)} = e_k \cdot \text{Down}(\mathbf{x}_1, 2^k) + (1 - e_k) \cdot \mathbf{x}_0^{(k)}$
12:        Compute the midpoints as $\mathbf{x}_t^{(k)} = \frac{e_k - t}{e_k - s_k} \cdot \mathbf{x}_s^{(k)} + \frac{t - s_k}{e_k - s_k} \cdot \mathbf{x}_e^{(k)}$
13:        Define the target as $\mathbf{u}_t(\mathbf{x}_t^{(k)} | \mathbf{x}_1) = \frac{\mathbf{x}_e^{(k)} - \mathbf{x}_s^{(k)}}{e_k - s_k}$
14:        Calculate loss as $\text{MSE}(\mathbf{u}_t(\mathbf{x}_t^{(k)} | \mathbf{x}_1), \mathbf{v}(\mathbf{x}_t^{(k)}))$
15:        Run back-propagation, update parameters of $\mathbf{v}$
16:     **end for**
17: **end for**
18: **Return:** $\mathbf{v}$

---

**Algorithm 4** Pyramidal Flow Sampling

---

1: **Input:** Trained flow model $\mathbf{v}$, number of stages $K$, list of $K$ start times $s_k$, total number of timesteps $N$.
2: **Output:** Generated image.
3: Sample the starting point $\hat{\mathbf{x}}_0^{K-1} \sim \mathcal{N}\left(0, I \cdot \frac{1}{4^{K-1}}\right)$
4: Sample Gaussian noise at maximum resolution $\mathbf{x}_0 \sim \mathcal{N}(0, I)$
5: **for** stage $k = K - 1$ to $0$ **do**
6:     $\hat{\mathbf{x}}_{s_{k-1}}^{(k-1)} = \text{ODEINT}(\mathbf{v}_t^{(k)}, \{s_k, s_{k-1}\}, \hat{\mathbf{x}}_{s_k}^{(k)})$
7:     Define the re-noising factor $n_k = (1 - s_{k-1})\left(\text{Down}(\hat{\mathbf{x}}_0, 2^{k-1}) - \text{Up}(\text{Down}(\hat{\mathbf{x}}_0, 2^k))\right)$
8:     Define the next starting point $\hat{\mathbf{x}}^{(k-1)} = \text{Up}(\hat{\mathbf{x}}^{(k)}) + n_k$
9: **end for**
10: **Return:** $\mathbf{x}_{K-1}$

---

---

**Algorithm 5** EdifyImage Flow Matching Training

---

1: **Input:** Dataset of images $D$, number of epochs $M$, critical time points $T_1$ and $T_2$, where $0 < T_2 < T_1 < 1$
2: **Output:** Trained flow model $\mathbf{v}_t^{(0)}, \mathbf{v}_t^{(1)}, \mathbf{v}_t^{(2)}$
3: Initialize the model weights
4: **for** epoch = 1 to $M$ **do**
5:    **for** $\mathbf{x}_1$ in $D$ **do**
6:       Sample the stage $k$ from a uniform distribution over $\{0, 1, 2\}$
7:       Sample $t$ from a uniform distribution over $[T_{k+1}, 1]$ where $T_0 = 1$ and $T_3 = 0$
8:       Sample the start point as random Gaussian $\mathbf{x}_0^{(0)} \sim \mathcal{N}(0, I^{(0)})$
9:       Compute noisy image $\mathbf{x}_t^{(k)} = (1-t)\text{Down}(\mathbf{x}_0^{(0)}, 2^k) + \mu^{(k)}(\mathbf{x}_1, t)$
10:      Compute velocity target $\mathbf{u}_t^{(k)}(\mathbf{x}_t^{(k)}|\mathbf{x}_1) = -\text{Down}(\mathbf{x}_0^{(0)}, 2^k) + \frac{d\mu^{(k)}(\mathbf{x}_1, t)}{dt}$
11:      Forward the model to get $\mathbf{v}_t^{(k)}(\mathbf{x}_t^{(k)})$
12:      Calculate loss MSE $\left( \mathbf{v}_t^{(k)}(\mathbf{x}_t^{(k)}), \mathbf{u}_t^{(k)}(\mathbf{x}_t^{(k)}|\mathbf{x}_1) \right)$
13:      Run back-propagation, update network parameters
14:    **end for**
15: **end for**
16: **Return:** Trained flow model $\mathbf{v}_t^{(2)}, \mathbf{v}_t^{(1)}, \mathbf{v}_t^{(0)}$

---

**Algorithm 6** EdifyImage Flow Matching Sampling

---

1: **Input:** Trained flow model $\mathbf{v}_t^{(0)}, \mathbf{v}_t^{(1)}, \mathbf{v}_t^{(2)}$, standard Gaussian noise of the largest scale $\mathbf{x}_0^{(0)}$, critical time points $T_1$ and $T_2$
2: **Output:** Largest resolution sample $\hat{\mathbf{x}}_1^{(0)}$
3: $\hat{\mathbf{x}}_{T_2}^{(2)} = \text{ODEINT}(\mathbf{v}_t^{(2)}, \{0, T_2\}, \text{Down}(\mathbf{x}_0^{(0)}, 4))$
4: $\hat{\mathbf{x}}_{T_2}^{(1)} = \text{Up}(\hat{\mathbf{x}}_{T_2}^{(2)}) + (1 - T_2)(\text{Down}(\mathbf{x}_0^{(0)}, 2) - \text{Up}(\text{Down}(\mathbf{x}_0^{(0)}, 4)))$ # This is the re-noise step
5: $\hat{\mathbf{x}}_{T_1}^{(1)} = \text{ODEINT}(\mathbf{v}_t^{(1)}, \{T_2, T_1\}, \hat{\mathbf{x}}_{T_2}^{(1)})$
6: $\hat{\mathbf{x}}_{T_1}^{(0)} = \text{Up}(\hat{\mathbf{x}}_{T_1}^{(1)}) + (1 - T_1)(\mathbf{x}_0^{(0)} - \text{Up}(\text{Down}(\mathbf{x}_0^{(0)}, 2))$
7: $\hat{\mathbf{x}}_1^{(0)} = \text{ODEINT}(\mathbf{v}_t^{(0)}, \{T_1, 1\}, \hat{\mathbf{x}}_{T_1}^{(0)})$
8: **Return:** $\hat{\mathbf{x}}_1^{(0)}$

---

## F.1 DISCUSSIONS WITH PIXELFLOW (CHEN ET AL., 2025)

Here we discuss the relationship between the concurrent work PixelFlow (Chen et al., 2025) and PyramidalFlow (Jin et al., 2025). The algorithm of PyramidalFlow is shown in Algorithm 3. Line 10 and Line 11 are exactly the same as Equation (2-3) of the PixelFlow paper, while Line 12 is the same as Equation (4) of PixelFlow. PixelFlow also uses a re-noising procedure as in PyramidalFlow, although details are not presented in their paper.

## F.2 SAMPLING IN PYRAMID FLOW MATCHING

Sampling from a multiscale Pyramidal Flow model is sequential, starting from stage $K - 1$ and ending at stage 0, where $K$ is the number of scales used in training. For each stage, two timesteps are defined: the start time $s_k$ and the end time $e_k$. In this section, we assume the stages to have zero time overlap, that is, $s_{k-1} = e_k$. Also, the stages together span the whole time interval, that is, $s_{K-1} = 0$ and $e_0 = 1$. The noise is sampled once in the largest dimension $n \sim \mathcal{N}(0, I)$. Within each stage the sampling process follows the standard flow path between $\mathbf{x}_s = s_{k-1} \cdot \text{Up}(\text{Down}(\mathbf{x}, 2^k)) + (1 - s_{k-1}) \cdot \text{Down}(n, 2^{k-1})$, and $\mathbf{x}_e = e_{k-1} \cdot \text{Down}(\mathbf{x}, 2^{k-1}) + (1 - e_{k-1}) \cdot \text{Down}(n, 2^{k-1})$.

The jump between consecutive stages needs to be handled carefully to enforce the consistency of the flow. Consider the jump from stage $k$ to stage $k - 1$. The procedure starts with up-sampling the

Table 6: **Training details across datasets and resolutions.** We summarize hyperparameters, computation complexity, and resource usage across benchmarks. Training time and speed may fluctuate due to hardware conditions.

| Training Details | CelebA-HQ 256 | CelebA-HQ 512 | CelebA-HQ 1024 | ImageNet 256 |
|---|---|---|---|---|
| Backbone | DiT-L/2 | DiT-L/2 | DiT-L/2 | DiT-B/2 / DiT-XL/2 |
| Largest latent size | $32 \times 32$ | $64 \times 64$ | $128 \times 128$ | $32 \times 32$ |
| Batch size | 256 | 256 | 128 | 256 |
| Init LR | 2e-4 | 2e-4 | 2e-4 | 1e-4 |
| LR schedule | CosineAnnealingLR | CosineAnnealingLR | CosineAnnealingLR | N/A |
| Final LR | 1e-6 | 1e-6 | 1e-6 | N/A |
| Optimizer | AdamW | AdamW | AdamW | AdamW |
| Training epochs | 500 | 1000 | 2000 | 1400/120 |
| Time segments | 0.5 | 0.33, 0.67 | 0.33, 0.67 | 0.5 |
| Number of scales | 2 | 3 | 3 | 2 |
| VAE type | EQVAE | SDVAE | SDVAE | EQVAE |
| EMA decay rate | N/A | N/A | N/A | 0.9999 |
| Hardware | $8\times$H200 | $8\times$H200 | $8\times$H200 | $8\times$H200 |
| Peak memory per GPU (GB) | 42.2 | 82.8 | 120.5 | 24.9 / 68.3 |
| Training wall-clock (days) | 1.2 | 3.1 | 8.3 | 8.5 / 9.4 |

endpoint of stage $k$, yielding

$$\mathrm{Up}(\mathbf{x}_e) = s_{k-1}\mathrm{Up}(\mathrm{Down}(x, 2^k)) + (1 - s_k)\mathrm{Up}(\mathrm{Down}(n, 2^k))$$

The distribution of this data differs from the distribution of the startpoint $\mathbf{x}_s^{k-1}$ of the $k-1$ stage.

$\mathrm{Down}(x, 2^{k-1}) \sim \mathcal{N}\left(0, I \cdot \frac{1}{2^{k-1}}\right)$, and similarly $\mathrm{Down}(x, 2^k) \sim \mathcal{N}\left(0, I \cdot \frac{1}{2^k}\right)$, however the upsampled noise $\mathrm{Up}(\mathrm{Down}(n, 2^k)) \sim \mathcal{N}\left(0, \Sigma \cdot \frac{1}{2^k}\right)$ where $\Sigma$ is a block-diagonal matrix that has $2 \times 2$ blocks of ones on the diagonal and zeroes elsewhere. Then if we define a different block-diagonal matrix $\Sigma'$ with blocks

$$\Sigma'_{block} = \begin{pmatrix} 1 & -1 \\ -1 & 1 \end{pmatrix}$$

and set $\alpha = \sqrt{\frac{3}{4^k}}$, and sample $m \sim \mathcal{N}(0, \Sigma')$, we have $Var(\mathrm{Up}(\mathbf{x}_e^k) + (1 - s_{k-1})\alpha \cdot m) = Var(\mathbf{x}_s^{k-1})$. In practice this means that at the end of stage $k$ we sample $m \sim \mathcal{N}(0, \Sigma')$ and set $x_s^{k-1} = \mathrm{Up}(\mathbf{x}_e^k) + (1 - s_{k-1})\alpha \cdot m$ as the starting point of stage $k-1$.

## G  MORE TRAINING DETAILS

We present the detailed training settings in Table 6, highlighting key implementation differences across datasets and resolutions. For CelebA-HQ, we utilize a DiT-L/2 backbone across all resolutions while adjusting the number of scales based on complexitytwo scales for $256 \times 256$ and three scales for higher resolutions. For ImageNet, we experiment with both DiT-B/2 and DiT-XL/2 backbones to demonstrate scaling properties. Notably, we use EQVAE for our lower-resolution models due to its superior reconstruction quality, while higher-resolution CelebA-HQ models leverage SDVAE for better stability. All models were trained on 8ŒH200 GPUs, with batch sizes adjusted according to memory constraints.

## H  LIMITATIONS

First, we didn't train our XL model on ImageNet for $7M$ steps due to a restriction of computational resources. This limitation may have prevented our model from reaching its full potential in comparison to diffusion models that are typically trained for significantly longer periods. Second, our approach still requires storing part of the model weights for each scale, which increases memory requirements during training, although our MoT design partially mitigates this issue. Third, while our method shows excellent performance on unconditional and conditional generation tasks, we have not yet

extended it to conditional generation scenarios such as text-to-image synthesis or inpainting, which are important applications for generative models. Finally, our method inherits limitations from the VAE architecture used for latent space projection, potentially limiting the fidelity of fine details in generated images. Future work could address these limitations by exploring more efficient parameter-sharing techniques across scales, developing conditional variants of our approach, implementing memory-efficient sampling strategies, and investigating improved latent space representations.

## I  IMPACT STATEMENT

This paper presents Laplacian Multi-scale Flow Matching (`LapFlow`), a novel approach for generative image modeling. We believe this work has several potential positive impacts on the research community and society:

Our method improves computational efficiency in high-resolution image generation, potentially reducing the energy consumption and carbon footprint associated with training and deploying generative models. By requiring fewer function evaluations and reduced GFLOPs, `LapFlow` contributes to more sustainable AI development.

The technical innovations in multi-scale representations and unified transformer architectures may inspire new approaches to other generative tasks beyond images, such as audio, video, and 3D content creation. The proposed mixture-of-transformers architecture with parameter sharing could influence more efficient model designs across various domains.

However, we acknowledge that advances in generative image modeling also come with potential societal concerns. Like other generative models, this technology could be misused to create misleading or deceptive content. We encourage responsible development and deployment of systems based on our research, including content safeguards and transparent disclosure of AI-generated media.

Our work focuses on advancing the technical capabilities of generative models rather than specific applications. We emphasize that any practical implementation should consider ethical implications and be developed in accordance with responsible AI principles.

