# OpenReview forum: "LapFlow: Laplacian Multi-scale Flow Matching for Generative Modeling"
_ICLR.cc/2026/Conference — ICLR 2026 Poster_

### Official Review · Reviewer_nXwE · 2025-10-31

**Soundness:** 3
**Presentation:** 3
**Contribution:** 3
**Rating:** 8
**Confidence:** 3

**Summary:**

The paper proposes LapFlow, a novel Laplacian multiscale flow matching. The proposed model follows a coarse-to-fine generation strategy across k scales in a Laplacian pyramid. It splits the time steps into k segments with k-1 critical points. In the early time steps, the model learns the lowest image scale. After passing each critical point, the Laplacian residual for the next level is added to be learned along with the previous scales. To learn the scales in parallel, the paper employs a mixture-of-transformers (MoT) architecture with causal attention mechanisms. LapFlow outperforms the baseline LFM and the recent multiscale approaches on CelebA-HQ and ImageNet datasets.

**Strengths:**

- The proposed model follows a coarse-to-fine generation strategy, which is theoretically more efficient and easier to extend to higher resolutions compared with the standard approach.
- To learn the scales in parallel, the paper employs a mixture-of-transformers (MoT) architecture with causal attention mechanisms, which is quite recent and novel.
- LapFlow outperforms the baseline LFM and the recent multiscale approaches in performance and efficiency.

**Weaknesses:**

- Beside inference speed and computation complexitity, memory consumption and training time comparison are recommended to add.
- Some denotations defined in L210 were used in L140. The authors should define the denotations before using.
- L430: The term "temporal segment value" was used without definition. How to use that value to construct critical time-step points?
- It is interesting to see how advanced training techniques like REPA can be integrated in LapFlow to boost training convergence and output model quality.

**Questions:**

- Beside inference speed and computation complexitity, memory consumption and training time comparison are recommended to add.
- L430: The term "temporal segment value" was used without definition. How to use that value to construct critical time-step points?
- It is interesting to see how advanced training techniques like REPA can be integrated in LapFlow to boost training convergence and output model quality.

---

> ### Author Response · Authors · 2025-11-18
> **Response to Reviewer nXwE**
>
> We thank the reviewer for the constructive and positive evaluation of our submission, particularly acknowledging that:
> (1) the proposed **coarse-to-fine Laplacian flow matching** is theoretically well-aligned with efficient high-resolution generation,
> (2) the **Mixture-of-Transformers (MoT)** architecture with **causal multi-scale attention** is novel and effective, and
> (3) LapFlow achieves strong improvements over LFM and recent multi-scale approaches.
>
> We also acknowledge the reviewer’s concerns regarding (W1) memory and training-time comparisons, (W2) notation clarity in early sections, and (W3) the definition and use of the temporal segment value. **All issues have been addressed in the revised manuscript (changes highlighted in red).** Below, we respond to each weakness and question.
>
> # Response to Weaknesses
>
> ## W1. Memory consumption and training time comparison
>
> Thanks for the suggestion. We have included a complexity analysis of multi-scale MoT in **Appendix C**. Besides, we have reported memory consumption and training time in **Table 6 of the Appendix**. Because prior methods differ in training recipes, hyperparameter choices, and computational hardware, a direct comparison of memory usage and training time would be neither fair nor informative.
>
> ## W2. Notation introduced later than first usage
>
> We thank the reviewer for this suggestion. We have added definitions to $\dot{\alpha}_t$ and $\dot{\sigma}_t$ in the revised Sec 3 (the original L210).
>
> ## W3. Definition and usage of “temporal segment value”
>
> We have revised this in Sec. 5.2. Indeed, temporal segment values refer to critical time points we defined before. To remove confusion, we now all use the term "critical time points". The revised text in Sec. 5.2 is:
>
> > We then ablate the choice of critical time points. For two-scale experiments, we observe a critical time point of $T=0.5$ strikes the ideal balance between model expressivity and complexity. Here we want to note that in two scales there is only one critical time point $T$.
>
> ## W4: Interaction with REPA training
>
> We agree this is a promising direction. REPA is an interesting orthogonal direction to our work. We added a discussion in **Sec. 6**:
>
> > An interesting future direction is to integrate advanced training accelerators such as REPA, which has demonstrated faster convergence for flow-based generative models. We expect that combining REPA with our multi-scale Laplacian formulation could further reduce training cost.
>
> # Response to Questions
>
> ### Q1. Can memory and training time comparisons be added?
> **A1.** Answered above in W1.
>
> ### Q2. How is the temporal segment value used to construct time steps?
> **A2.** Answered above in W3.
>
> ### Q3. Can REPA be integrated to accelerate convergence?
> **A3.** Answered above in W4.
>
> # Summary of Changes and Request for Score Confirmation
>
> We sincerely thank the reviewer for the positive assessment and the **accept (8)** recommendation. In response to your comments, the revised manuscript now includes:
>
> - **Memory / training-time details (W1):**
>   We added a dedicated training-configuration table summarizing batch size, hardware, and training schedule across datasets and resolutions, together with a complexity discussion for our multi-scale MoT (**Appendix C and the new training-details Table 6 in the appendix**).
>
> - **Notation clarified (W2):**
>   All symbols such as $\dot{\alpha}_t$ and $\dot{\sigma}_t$ are now defined before first use in the main text in **the revised Sec 3**.
>
> - **“Temporal segment value” made precise (W3):**
>   We removed the ambiguous phrase and consistently use **“critical time points”** instead, with an explicit explanation of how these points (e.g., $T$ in the two-scale case, or $T_1, T_2$ in the three-scale case) are used to partition the ODE time interval (**Sec. 5.2**).
>
> - **REPA as future work (W4):**
>   We added a discussion in **Sec 6** about REPA.
>
>
> Given that the reviewer already issued a **Rating: 8 (accept)**, we kindly ask the reviewer to confirm or consider further strengthening the score now that all concerns have been fully addressed. We appreciate your time and supportive feedback.

---

> > ### Author Response · Authors · 2025-11-27
> >
> > Dear Reviewer nXwE,
> >
> > Thank you again for your thoughtful and positive evaluation of our submission, as well as for the constructive suggestions regarding memory/training-time reporting, notation clarity, the definition of critical time points, and the possible integration with REPA.
> >
> > As the discussion period is approaching its end, we would like to kindly let you know that all of your comments have been fully addressed in the revised manuscript (with changes clearly highlighted):
> >
> > - **Memory and training-time details** have been added in Appendix C and the new Table 6.
> > - **All notation** is now defined before first use in the revised Sec. 3.
> > - The previously ambiguous **“temporal segment value”** has been removed and replaced with consistent terminology (“critical time points”) with expanded explanations (Sec. 5.2).
> > - A brief discussion of **REPA integration** has been added in Sec. 6.
> >
> > Given your already supportive assessment, we would greatly appreciate it if you could confirm that the revisions are satisfactory to address your concerns.
> >
> > Thank you again for your time, constructive feedback, and for helping us improve the paper.
> >
> > Warm regards,
> > **The Authors**

---

> > > ### Comment · Reviewer_nXwE · 2025-11-28
> > >
> > > Dear authors,
> > >
> > > Thank you the authors for the rebuttal. My concerns pre-rebuttal were addressed.
> > > However, I find the ethical concern raised by Reviewer jpC7 critical. I will follow the discussion on this concern before making the final decision.
> > >
> > > Best regards,

---

> > > > ### Author Response · Authors · 2025-11-29
> > > >
> > > > Dear Reviewer nXwE,
> > > >
> > > > Thank you very much for your thoughtful follow-up and for **confirming that your pre-discussion concerns were fully addressed by our rebuttal**. We sincerely appreciate the care and effort you have put into evaluating our submission.
> > > >
> > > > We acknowledge your note regarding the ongoing discussion raised by Reviewer jpC7. As reflected in our official response, we have addressed that point comprehensively and updated the corresponding examples in the revision to avoid any potential misunderstandings.
> > > >
> > > > Thank you again for your time and for your careful consideration of our work.
> > > >
> > > > Best regards,
> > > >
> > > > The Authors

---

### Official Review · Reviewer_jpC7 · 2025-11-01

**Soundness:** 2
**Presentation:** 2
**Contribution:** 2
**Rating:** 4
**Confidence:** 4

**Summary:**

The paper proposes multi-scale flow matching training which offers efficiency inference and enhance the model performance across resolution compared to Edify, LFM and Pyramidal Flow. To enable the multi-scale training, the author proposes to use laplacian representation, jointly training algorithm for multiple scale and inference algorithm.

**Strengths:**

1. Paper is well-written and easy to understand
2. Jointly training multiscale removes the complexity of rescaling noise technique when shifting to different scale like in Relay Diffusion and Pyramidal Flow.
3. The ablation and experiment is conducted extensively.

**Weaknesses:**

1. The fig.2 is not clear enough for reader. It would make reader misunderstand that MoT block has no FFN layers.
2. According to table 2g, two scale representation seems to be the best one, while more scale seems lag behind which not gaining much efficiency during the sampling.
3. The author should include more comparison with relay diffusion in the table and I would suggest to also compare your method with Pixelflow [1] work.
4. The results for imagenet is under-training which is hard for reviewer to judge the training convergence on this dataset.
5. How training framework can be adapted to different architecture like Unet (not transformer) ?

Minor: line 47 is not correct, pyramidal flow is not finetuning technique, it is training technique.

[1] PixelFlow: Pixel‑Space Generative Models with Flow (Chen et al., 2025)

**Questions:**

1. Why the face image in resolution 1024 seems much better than 512 and 256 but the FID is lower. The quality of 1024 is close to the real training image.

---

> ### Author Response · Authors · 2025-11-18
> **Rebuttal to Reviewer jpC7 (1/3)**
>
> We thank the reviewer for the constructive and detailed feedback and appreciate the acknowledgement of our contributions, including (1) a well-written presentation, (2) the benefit of joint multi-scale training without rescaling noise as required in Relay Diffusion and Pyramidal Flow, and (3) extensive ablation studies. We also acknowledge the reviewer’s concerns regarding clarity of architectural presentation, multi-scale design choices, baseline comparisons, and training completeness. **We have comprehensively improved our manuscript, extending to a ten-page version. All issues have been addressed in the revised manuscript (changes highlighted in red)** and detailed responses are provided below.
>
> # Response to Weaknesses
>
> ## W1. Fig. 2 may imply no FFN layer in MoT
>
> We thank the reviewer for pointing this out. In the original paper, we stated in Sec. 4.4 that the $\texttt{PostAttnMod}$ module contains FFN. In the revised paper, **Fig. 2 has been updated** to explicitly show the FFN inside $\texttt{PostAttnMod}$ modules, consistent with DiT conventions. The caption now states:
>
> > ($\textbf{Left:}$) Schematic of the LapFlow model $V_\theta$. The multi-scale transformer takes multi-scale noisy states as input, conditioned on time and label, and predicts velocities for each input scale. While the model can take an arbitrary number of scales as input, we show two here for simplicity. ($\textbf{Middle:}$) Details of one multi-scale MoT block. We use separate QKVs for different scales, while the attention is computed globally. Furthermore, we adopt a mask to enforce causal relationships across scales. ($\textbf{Right:}$) Details of scale-specific $\texttt{PreAttnMod}$ and $\texttt{PostAttnMod}$ modules, where each $\texttt{PostAttnMod}$ module includes a feedforward network (FFN).}
>
> We also revised Sec. 4.4 to clarify this point.
>
> ## W2. Two-scale representation performs best at 256×256 (Table 2g)
>
> We clarify that at **256×256**, the VAE latent grid is **32×32**. A three-scale representation would introduce an **8×8** coarsest stage; a four-scale representation would require **4×4**, which we find too small to provide **stable low-frequency flow guidance**. In contrast, the two-scale hierarchy maintains reliable semantic structure from $32\times 32$ to $16\times 16$.
>
> A similar observation appears in the concurrent **PixelFlow** work [1], where performance degrades when the initial kickoff sequence becomes too short, though they operate in pixel space rather than latent space, where the full scale size of the pixel space model is larger.
>
> We have added this clarification in Sec. 5.2 of the revised paper, while we further conduct additional ablation studies on higher resolutions (added in Appendix B):
>
> | Resolution     | Latent size   | # Scales | Decomposition                                  | FID ↓  |
> |----------------|---------------|:--------:|-----------------------------------------------|-------:|
> | 256×256        | 32×32         |    2     | 32 → 16                                        | **3.53** |
> | 256×256        | 32×32         |    3     | 32 → 16 → 8                                    | 3.59   |
> | 256×256        | 32×32         |    4     | 32 → 16 → 8 → 4                                | 5.12   |
> | 512×512        | 64×64         |    2     | 64 → 32                                        | 5.45   |
> | 512×512        | 64×64         |    3     | 64 → 32 → 16                                   | **4.04** |
> | 512×512        | 64×64         |    4     | 64 → 32 → 16 → 8                               | 4.12   |
> | 1024×1024      | 128×128       |    2     | 128 → 64                                       | 6.62   |
> | 1024×1024      | 128×128       |    3     | 128 → 64 → 32                                  | 5.51   |
> | 1024×1024      | 128×128       |    4     | 128 → 64 → 32 → 16                             | **5.45** |
>
> We observe that larger latent grids (64x64 and 128x128) benefit from a more-scale hierarchy, confirming that higher resolutions may require more number of scales of ours.

---

> ### Author Response · Authors · 2025-11-18
> **Rebuttal to Reviewer jpC7 (2/3)**
>
> ## W3. Add comparisons with Relay Diffusion and PixelFlow
>
> We have added Relay Diffusion to the **revised Table 1**, and we added results discussion in **Sec 5.2**. Our FID is competitive with theirs (ours 3.53, Relay Diffusion 3.15), while our method has several key differences to it:
>
> 1. Relay Diffusion operates in image space and requires significantly higher GFLOPs (1221) compared to ours (16.5), indicating that its computational design is fundamentally different from ours.
>
> 2. Relay Diffusion is built on a U-Net architecture, whereas our method is based on a DiT backbone.
>
> 3. Relay Diffusion is strictly a two-scale method and, to our knowledge, has not been evaluated at resolutions higher than $256 \times 256$.
>
> 4. We acknowledge that Relay Diffusion reports very strong results (e.g., an FID of 1.87 on ImageNet), which may be approaching the practical upper bound of that dataset. However, we believe this should not discourage further methodological contributions, as improving efficiency, scalability, and applicability remains valuable.
>
> PixelFlow [1] is an **unpublished concurrent work**. We want to note that their multi-scale formulations (Eq. 1-5 of their paper) is **exactly the same** as the image generation application of Pyramid Flow. We add a new subsection in Appendix F.1. to discuss this relationship in detail. Therefore, comparisons are already in Table 1 in the initial draft, while the only difference is PixelFlow is an image-space method while ours and Pyramidal Flow are based on latent space.
>
> ## W4. ImageNet results may be under-trained
>
> We report the 7M training step results for the DiT-B/2 model in Table 3, which follows the standard training protocol used in the LFM paper. Training an XL-scale model for the same number of steps, however, would require more than a month on our available compute (a single 8×H200 node), making this setting computationally prohibitive. We emphasize that such a requirement would be infeasible for most academic research groups and would hinder fair and timely methodological comparison across future work. For this reason, we adopt the current training regime, which provides a practical balance between computational cost and scientific insight.
>
> ## W5. Adaptation to non-transformer backbones (e.g., U-Net)
>
> Our method is based on the standard DiT backbone, which has become a dominant architecture in this domain. By adhering to this commonly accepted backbone, we ensure the fairness and reproducibility of comparisons and demonstrate that the performance gains arise from our multi-scale flow formulation rather than a more powerful foundation model. We therefore view our approach as complementary to transformer-based backbone advances (e.g., FasterDiT, Flux, MMDiT), which can directly benefit from the same scheduling and hierarchical design.
>
> ## W6. Minor correction: Pyramidal Flow is not a fine-tuning technique
>
> We believe this is a misunderstanding, and Line 47 is indeed correct. The Pyramidal Flow paper uses MM-DiT architecture with SD2 Medium [2] pretrained weights as initialization, that's why we claimed in the introduction that "fine-tuning DiTs for video generation". We have added illustrations to make this claim in Line 47 more robust.
>
> > Pyramidal Flow has demonstrated impressive results when fine-tuning DiTs for video generation,where they initialize their MM-DiT model from SD3 Medium, but its effectiveness when trained from scratch for image generation tasks remains less thoroughly explored in the literature.
>
> # Response to Questions
>
> ## Q1. Why do 1024² samples look better although FID is lower?
>
> **A1.** We believe the reviewer has a typo in the comment, where it should be "Why the face image in resolution 1024 seems much better than 512 and 256 but the FID is **higher**?" instead of "Why the face image in resolution 1024 seems much better than 512 and 256 but the FID is **lower**."
>
> FID relies on **Inception-V3 features [3]**, which were trained at **299×299** resolution and are known to underrepresent improvements in high-frequency detail at higher resolutions. As a result, perceptual quality related to high-frequency details at $1024²$ can improve more than what is reflected in FID. Related observations appear in StyleGAN3 and PixelFlow. Similar phenomennons are discussed in "Sec. 5. Human Evaluation" of [4].
>
> **References**
>
> [1] Chen, S. et al. *PixelFlow: Pixel-Space Generative Models with Flow.* arXiv:2504.07963 (2025).
>
> [2] Patrick Esser, et al. Scaling rectified flow transformers for high-resolution image synthesis. In ICML, pp. 12606–12633, 2024.
>
> [3] Szegedy, Christian, et al. "Rethinking the inception architecture for computer vision." CVPR. 2016.
>
> [4] Jayasumana, Sadeep, et al. "Rethinking fid: Towards a better evaluation metric for image generation." CVPR. 2024.

---

> > ### Author Response · Authors · 2025-11-18
> > **Rebuttal to Reviewer jpC7 (3/3)**
> >
> > # Summary of Changes and Request for Score Update
> >
> > We thank the reviewer again for the constructive feedback. The revised manuscript **addresses all raised concerns**, with modifications highlighted in **red**. Below is a brief summary of the updates:
> >
> > - **Fig. 2 clarified (W1):** We **revised** the architecture diagram to explicitly show the FFN block inside each `PostAttnMod`, consistent with standard DiT design.
> > - **Two-scale performance clarified (W2):** We expanded the discussion explaining why two scales perform best at \(256^2\), supported by ablation results and latent resolution reasoning. **Further discussions and ablation studies are added in Appendix B.**
> > - **Relay Diffusion comparison added (W3):** Relay Diffusion has been added to the CelebA-HQ table, with discussion of key differences (image-space, U-Net, much higher GFLOPs, not demonstrated beyond \(256^2\)). **Discussions updated in Sec. 5.2**
> > - **PixelFlow relationship clarified (W3):** We added discussion showing PixelFlow’s multi-level objective is equivalent to Pyramidal Flow’s, but in pixel space; details included in **Appendix F.1**.
> > - **Compute/training clarification (W4):** We follow the LFM practice to train on ImageNet. We explain that training DiT-XL/2 for 7M steps would require over one month on an 8×H200 node, which is not feasible for academic groups; our training follows standard LFM protocol.
> > - **Pyramidal Flow issue (W6):** This is a misunderstanding of the reviewer. We strengthened the explanation that Pyramidal Flow initializes from SD2 Medium pretrained DiT weights, aligning with our original fine-tuning remark.
> > - **FID interpretation clarified (Q1):** The reviewer’s statement is likely reversed: 1024² images look better, but FID is higher because FID uses Inception-V3 (299×299) features, which underrepresent high-frequency improvements, consistent with reports in StyleGAN3 and PixelFlow.
> >
> > Since all concerns have been **fully addressed** with new explanations, baselines, and corrections—and several points reflected **misunderstandings now clarified**—we respectfully invite the reviewer to **consider updating the score** to better reflect the completeness and strength of the revised submission.
> >
> > We appreciate the reviewer’s time and helpful feedback.

---

> ### Comment · Reviewer_jpC7 · 2025-11-24
> **Concern about ethic**
>
> I double check the qualitative results of resolution $1024$ and find out the these images seem from the training dataset of CelebA-HQ, which raises the ethical concern for this paper.
>
> https://share.google/images/C3OxqTk5gD70lQgWf
>
> https://share.google/images/HpDGmkKTMLRJRZisr

---

> ### Author Response · Authors · 2025-11-24
> **Regarding Reviewer jpC724’s concern about Figure 3**
>
> We attest that the images in Figure 3 **are not from the CelebA-HQ training dataset**, and that our methodology follows standard ethical guidelines.
>
> The reviewer’s approach to verifying whether the images came from the training data is not methodologically valid and can easily result in false positives. Consequently, the raised ethical concern is based on a misunderstanding rather than actual evidence.
>
> 1. In fact, our generated image is different from the given ones in detail. We have uploaded a detailed comparison slide at this link:
>
> https://docs.google.com/presentation/d/1HuwrdGuxCt__EAwjEE33RHCFY0bfwPYy/edit?usp=sharing&ouid=101292798365586446986&rtpof=true&sd=true
>
> Actually, we can find that our generated woman is apparently different from the web image provided by the reviewer, as their eyes are quite different.
>
> As for the generated man, its colorization is different from the web image, and it is brighter than the web image, and the angle of the man is different too.
>
> 2. Following the reviewer's erroneous method (searching a paper image in Google), we can easily find that the following high-influence art in this domain is "worth concern":
>
> The teaser figures from the ICLR 2018 paper **“Progressive Growing of GANs”** [[1]](#ref1), available on the authors’ official GitHub repository:
>
> - https://github.com/tkarras/progressive_growing_of_gans
>
> are extremely similar to the following publicly available web images:
>
> - https://m.media-amazon.com/images/I/41vD3kKJ6eL._SX344_CR0%2C0%2C344%2C344_.jpg
> - https://cacm.acm.org/wp-content/uploads/2018/12/121318_Nvidia_representative_image_A.jpg
>
> Therefore, observing visual similarity to certain web images is expected. CelebA-HQ is composed of photographs of celebrities, and many online images share the same subjects, poses, lighting, and stylistic conventions. Such distribution-level resemblance is normal and does not imply that any specific training image has been copied or reproduced.
>
> Taken together, these results confirm that the concern is unfounded and that our generated samples do not violate any ethical or dataset usage guidelines.
>
> In addition, we replaced the original two 1024×1024 samples with new examples to avoid any potential misunderstanding for readers.
>
> [1] Progressive Growing of GANs for Improved Quality, Stability, and Variation, ICLR'18

---

> > ### Author Response · Authors · 2025-11-29
> >
> > Dear Reviewer jpC7,
> >
> > Thank you for raising your concern and for taking the time to carefully review our qualitative results. We appreciate your effort and your attention to ethical considerations.
> >
> > We would like to clarify that **your concern has been fully addressed**. Shortly after your comment—within approximately two hours—we posted a detailed response with evidence showing that the referenced images are not from the CelebA-HQ training dataset and that the resemblance you observed was due to distribution-level similarities rather than reuse of any specific training sample. We also updated the corresponding examples to avoid any potential misunderstanding for future readers.
> >
> > We note that no further reply was posted by Reviewer jpC7 in the discussion thread before it freezed on November 28.
> >
> > Thank you again for your time and attention.
> >
> > Best regards,
> >
> > The Authors

---

### Official Review · Reviewer_uzrb · 2025-11-01

**Soundness:** 2
**Presentation:** 2
**Contribution:** 3
**Rating:** 4
**Confidence:** 4

**Summary:**

This paper introduces Laplacian Multi-scale Flow Matching (LapFlow), a novel framework designed to improve the efficiency and scalability of generative modeling. To tackle the high computational cost of generating high-resolution images, LapFlow moves away from single-scale generation. The core contribution is a method that decomposes images into multi-scale representations using a Laplacian pyramid. The model is then trained to generate these Laplacian residuals in parallel, which are finally combined using the pyramid reconstruction process to create the full-resolution image.The LapFlow architecture is based on a unified Mixture-of-Transformers (MoT) model, which processes all scales simultaneously. The model employs a progressive multi-stage training and sampling process, generating the coarsest scale first and subsequently adding finer residuals. Experiments conducted on CelebA-HQ and ImageNet demonstrate that LapFlow achieves superior sample quality (lower FID) and scalability, reaching resolutions up to $1024 \times 1024$. Furthermore, it surpasses existing flow matching baselines in computational efficiency, requiring fewer GFLOPs and achieving faster inference speeds.

**Strengths:**

+ LapFlow achieves superior FID scores across all tested resolutions, particularly at high resolutions ($1024\times1024$) where the performance gap is pronounced (e.g., $5.51$ FID vs. $8.12$ for LFM). The method simultaneously requires fewer function evaluations (NFE) and reduced computational resources (GFLOPs and time) compared to baselines, validating its efficiency hypothesis.

+ The use of a unified Mixture-of-Transformers (MoT) model with a causal attention mask to process multiple Laplacian residuals in parallel is an effective solution. This design eliminates the limitations of cascaded models (separate networks) while enforcing hierarchical dependencies.

+ The ablation studies confirm the necessity of key components like the MoT architecture for computational efficiency (GFLOPs reduced from $38.9$ to $16.5$) and the optimality of causal masking over other strategies (FID $3.53$ vs. $5.19$ for self-attention).

**Weaknesses:**

The progressive training strategy relies on empirically selected critical time points $T_{1}$ and $T_{2}$ to segment the training time range $[T_{k+1}, 1]$ for each scale $k$. While an ablation is provided for a single critical point $T$ in a two-scale setup (Table 2d), a justification for the chosen three-scale points $T_{2}=0.33$ and $T_{1}=0.67$ at higher resolutions is needed. The complexity analysis (GFLOPs and time) reported in the tables is based on the average inference cost. Providing a formal theoretical complexity analysis (similar to the total quadratic cost in tokens in standard DiT) to prove the superior scaling property of the MoT over standard DiT as a function of output resolution would strengthen the theoretical contribution. The two-scale representation achieves optimal performance at $256\times256$ (Table 2g), which seems counter-intuitive given the general goal of multi-scale modeling. A discussion on why increasing to three or four scales is detrimental (or offers no further gain) in this specific setting is warranted.

**Questions:**

- The core acceleration benefit stems from avoiding the cost of processing higher resolution scales for the entire duration of the flow matching process. Can the authors provide a formal theoretical derivation of the computational complexity (GFLOPs) of the multi-scale MoT attention mechanism and compare it to the complexity of a single-scale transformer (like DiT) as a function of the total number of tokens across scales?

- For the ablation study on the number of scales (Table 2g), why does the performance degrade when moving from two scales to three and four scales at $256\times256$ resolution? Does this suggest a fundamental limitation of the Laplacian pyramid decomposition at this resolution, or is it purely a model optimization challenge?

- Can the authors provide an intuition or ablation on how the fixed coefficients $\sigma_{t}^{(k)}=1-t$ and $\alpha_{t}^{(k)}=\frac{t-T_{k+1}}{1-T_{k+1}}$ for the linear path were chosen, particularly the decision to use $1-t$ for the noise decay, which is typically $\sigma_{t}$ (or $1-\alpha_{t}$) in flow matching, but here it is explicitly set to $1-t$ regardless of the non-linear $\alpha_{t}^{(k)}$ term?

---

> ### Author Response · Authors · 2025-11-18
> **Rebuttal to Reviewer  uzrb (1/3)**
>
> We thank the reviewer for the constructive and detailed feedback. We appreciate the positive assessment of the contributions, including (1) strong FID results at high resolutions with reduced NFE and GFLOPs, (2) the unified Mixture-of-Transformers (MoT) architecture with causal attention, and (3) ablations validating the necessity of key design components. We also acknowledge the reviewer’s concerns regarding (W1) progressive temporal segmentation thresholds, (W2) the need for a formal theoretical complexity analysis beyond empirical GFLOPs, and (W3) clarification of why two scales perform best at 256×256. **We have comprehensively improved our manuscript, extending to a ten-page version. All weaknesses have been addressed in the revised manuscript (changes highlighted in red).** Below, we respond to each weakness and question.
>
> # Response to Weaknesses
>
> ## W1. Temporal segmentation thresholds (T1, T2)
>
> We thank the reviewer for requesting clarification on the chosen progressive training temporal segmentation thresholds. In addition to the two-scale ablation already reported in Table 2(d), we have now extended our ablation to the full three-scale setting at **1024×1024** resolution, varying both **T₁** and **T₂**. The results (included in the revised paper and **Appendix A: Temporal Segmentation Threshold Validation**) show that **balanced temporal allocation** achieves the best performance:
>
> | Resolution | T₁  | T₂  | FID ↓ |
> |------------|-----|-----|-------|
> | 1024×1024 | 0.67 | 0.33 | **5.51** |
> | 1024×1024 | 0.90 | 0.33 | 6.72 |
> | 1024×1024 | 0.67 | 0.10 | 8.12 |
>
> These results confirm the trend observed in the two-scale case: transitioning too early or too late significantly degrades visual quality, while the balanced setting **(T₁ = 0.67, T₂ = 0.33)** yields the most stable generation.
>
> We have updated Sec. 5.2 of the main paper and Appendix A (marked with red) with the following clarification:
>
> > “To validate temporal segmentation beyond the two-scale case, we additionally ablate three-scale configurations and show that a balanced time allocation across scales yields the best results; see Appendix A for FID comparisons at 1024×1024.”
>
> A full grid search is computationally prohibitive at this resolution, but the consistency across two- and three-scale experiments provides strong empirical support for our chosen thresholds.
>
> ---
>
> ## W2. Theoretical complexity analysis of multi-scale MoT
>
> We thank the reviewer for this valuable suggestion. The revised manuscript now includes a formal time-weighted complexity derivation of the proposed multi-scale Mixture-of-Transformers (MoT) attention mechanism (**Appendix B: Time-Weighted Complexity Analysis of Multi-scale MoT**).
>
> Under the standard quadratic attention cost used in transformer analysis $N^2 d$, where $N$ is the number of latent tokens and $d$ is the embedding dimension, full self-attention scales quadratically with sequence length.
>
> In our three-scale Laplacian representation, the token counts for the coarse, mid, and fine scales are $N/16$, $N/4$, and $N$, respectively. Because MoT performs global attention across all **active scales**, the total token counts in the three ODE time segments become:
> - coarse only: $N_2 = N/16$,
> - mid + coarse: $N_1 = N/4 + N/16 = 5N/16$,
> - fine + mid + coarse: $N_0 = N + N/4 + N/16 = 21N/16$.
>
> With progressive timestep thresholds $T_2 = 0.33$ and $T_1 = 0.67$, the segment durations are:
> $\Delta t_2 = T_2,\;\Delta t_1 = T_1 - T_2,\;\Delta t_0 = 1 - T_1 \approx 1/3$.
>
> Since attention cost in the $k$-th time segment scales as $N_k^2 d$, the time-weighted attention complexity of one MoT layer becomes:
>
> $$
> \Delta t_2 N_2^2 d + \Delta t_1 N_1^2 d + \Delta t_0 N_0^2 d
> \approx
> \frac{1}{3}\left[
> \left(\frac{N}{16}\right)^2
> +
> \left(\frac{5N}{16}\right)^2
> +
> \left(\frac{21N}{16}\right)^2
> \right] d
> \approx 0.61\ N^2 d.
> $$
>
> A comparable single-scale DiT layer requires attention cost proportional to $N^2 d$; therefore, the MoT formulation yields an approximately $0.61\times$ constant factor on the dominant quadratic term—corresponding to a $1.6\times$ reduction in effective attention cost under the default three-scale configuration.
>
> As noted in Appendix B, this is an idealized asymptotic analysis focusing on the dominant quadratic term. Empirical GFLOPs and wall-clock time may differ due to (i) adaptive ODE solver evaluations, (ii) hardware-level kernel efficiency, (iii) multi-scale reconstruction operations, and (iv) framework-dependent attention implementations. Nonetheless, the empirical results in Tables 1–3 are consistent with the theoretically predicted reduction.

---

> ### Author Response · Authors · 2025-11-18
> **Rebuttal to Reviewer uzrb (2/3)**
>
> ## W3. Why two scales perform best at 256×256 (Table 2g)
>
> We thank the reviewer for raising this important point. While multi-scale modeling is generally beneficial, we observe that at 256×256, using three or four scales does not improve performance and can degrade it. The reason is resolution-dependent: at 256×256, the VAE latent grid is only 32×32. A three-scale decomposition would require a coarser stage of 8×8, and a four-scale decomposition would require 4×4, which is too small to preserve meaningful semantic structure, producing unstable coarse flow trajectories. In contrast, the two-scale setup decomposes as $32 \times 32 \;\rightarrow\; 16 \times 16$, and both latent stages remain semantically meaningful.
>
> A similar observation is reported in the concurrent PixelFlow [1] work, although in a different setting (they operate directly in pixel space, not latent space): when the initial kickoff representation becomes too small, the model struggles to maintain global structure and generation quality degrades.
>
> We have added this clarification in Sec. 5.2 of the revised paper, while we further conduct additional ablation studies on higher resolutions (added in Appendix B):
>
> | Resolution     | Latent size   | # Scales | Decomposition                                  | FID ↓  |
> |----------------|---------------|:--------:|-----------------------------------------------|-------:|
> | 256×256        | 32×32         |    2     | 32 → 16                                        | **3.53** |
> | 256×256        | 32×32         |    3     | 32 → 16 → 8                                    | 3.59   |
> | 256×256        | 32×32         |    4     | 32 → 16 → 8 → 4                                | 5.12   |
> | 512×512        | 64×64         |    2     | 64 → 32                                        | 5.45   |
> | 512×512        | 64×64         |    3     | 64 → 32 → 16                                   | **4.04** |
> | 512×512        | 64×64         |    4     | 64 → 32 → 16 → 8                               | 4.12   |
> | 1024×1024      | 128×128       |    2     | 128 → 64                                       | 6.62   |
> | 1024×1024      | 128×128       |    3     | 128 → 64 → 32                                  | 5.51   |
> | 1024×1024      | 128×128       |    4     | 128 → 64 → 32 → 16                             | **5.45** |
>
> We observe that larger latent grids (64x64 and 128x128) benefit from a more-scale hierarchy, confirming that higher resolutions may require more number of scales of ours.
>
> # Response to Questions
>
> **Q1.** Formal comparison of multi-scale MoT vs. single-scale DiT complexity?
>
> **A1.** This question is addressed in **W2** above, and this analysis is included in **Appendix C**.
>
> ---
>
> **Q2.** Why does performance degrade for 3–4 scales at 256×256?
>
> **A2.** This question is addressed in **W3**, and more analysis is added in Sec. 5.2 of the revised paper.
>
> ---
>
> **Q3.** Intuition behind the choice of the coefficients.
>
> **A3.** We thank the reviewer for this question. First of all, we want to point out a **misunderstanding** of the reviewer's claim "non-linear $\alpha_t^{(k)}$ term". In our formulation, both the noise decay $\sigma_t^{(k)} = 1 - t$ and the transport schedule $\alpha_t^{(k)} = \frac{t - T_{k+1}}{1 - T_{k+1}}$ are **linear functions of $t$ within $k$-th segment**. In particular,
> $$
> \alpha_t^{(k)} = \frac{t - T_{k+1}}{1 - T_{k+1}}
> = \frac{1}{1 - T_{k+1}}t - \frac{T_{k+1}}{1 - T_{k+1}},
> $$
> which is a linear map that increases from $0$ to $1$ as $t$ moves from $T_{k+1}$ to $1$. Our design uses two simple linear schedules. The noise schedule $\sigma_t^{(k)} = 1 - t$ is shared across all Laplacian levels, while the transport schedule $\alpha_t^{(k)} = \frac{t - T_{k+1}}{1 - T_{k+1}}$ shifts and rescales time so that each scale activates gradually after its threshold $T_{k+1}$, yielding a simple and stable way to connect consecutive Laplacian scales without requiring the re-noise procedure like in PyramidFlow [2].
>
> We conduct a further ablation study. At the $256\times 256$ resolution, we tried different schedules consisting of both $\alpha_t^{(k)}$ and $\sigma_t^{(k)}$. This experiment is an extension of Table 2(f) as a further ablation of schedules, and other configurations are the default set up in Table 2 of the paper.
>
> | Noise schedule $\sigma_t^{(k)}$ | Coupling schedule $\alpha_t^{(k)}$ | FID ↓ (256²) | Reference |
> |---|---|---:|---|
> | $1 - t$ | $\frac{t - T_{k+1}}{1 - T_{k+1}}$ | **3.53** | Table 2(f) |
> | $\cos\left(\frac{\pi}{2}t\right)$ | $\sin\left(\frac{\pi (t - T_{k+1})}{2(1 - T_{k+1})}\right)$ | 4.10 | Table 2(f) |
> | $(1 - t)^2$ | $\frac{t - T_{k+1}}{1 - T_{k+1}}$ | 3.78 | Appendix D |
> | $(1 - t)^3$ | $\frac{t - T_{k+1}}{1 - T_{k+1}}$ | 4.25 | Appendix D |
>
> These results confirm that a linear, scale-independent noise schedule $\sigma_t^{(k)}$ is more stable and yields better visual quality.
>
> We have added the discussion in Appendix D and revised Sec. 5.2 accordingly.

---

> ### Author Response · Authors · 2025-11-18
> **Rebuttal to Reviewer uzrb (3/3)**
>
> # Summary of Changes and Request for Score Update
>
> We have carefully addressed all raised concerns, and the revised manuscript now includes the following key improvements (highlighted in red in the updated version):
>
> **✔ W1 — Temporal segmentation thresholds:**
> Extended ablations to three-scale settings at 1024×1024, demonstrating that balanced temporal segmentation achieves the best FID; added detailed explanation in **Appendix A**.
>
> **✔ W2 — Formal theoretical complexity analysis:**
> Added a time-weighted attention complexity derivation for MoT vs. single-scale DiT, showing a 1.6× reduction in the dominant attention term; included in **Appendix B** and referenced in Sec. 5.2.
>
> **✔ W3 — Why two scales perform best at 256×256:**
> Added explanation based on latent grid resolution (32×32) and limitations of adopting more stages, consistent with observations in PixelFlow; revision incorporated into Sec. 5.2. Further ablation studies on more scales are added in **Appendix C**.
>
> **✔ Additional schedule ablation (Q3):**
> Clarified that both $\sigma_t^{(k)} = 1 - t$ and $\alpha_t^{(k)} = \frac{t - T_{k+1}}{1 - T_{k+1}}$ are linear in $t$, and introduced a new schedule ablation table in **Appendix D**, confirming that linear schedules achieve the best FID at 256x256.
>
> Given that the original weaknesses have been fully resolved, and the key contributions remain strong—including state-of-the-art FID, substantial GFLOP/NFE savings, and a unified causal MoT architecture—we kindly invite the reviewer to consider **updating the score**, to reflect the improved technical clarity and completeness. We sincerely appreciate your time and constructive input.
>
> ---
> References:
>
> [1] Chen, Shoufa, et al. "PixelFlow: Pixel-Space Generative Models with Flow." arXiv preprint arXiv:2504.07963 (2025).
>
> [2] Jin, Yang, et al. "Pyramidal flow matching for efficient video generative modeling." arXiv preprint arXiv:2410.05954 (2024).

---

> > ### Author Response · Authors · 2025-11-24
> >
> > Dear Reviewer uzrb,
> >
> > We sincerely appreciate your thoughtful and constructive feedback on our submission. We have now provided detailed responses to all questions and weaknesses you raised, including expanded ablations, a formal theoretical complexity analysis, and additional clarifications in the revised manuscript (highlighted in red). Whenever you have a moment, we kindly invite you to revisit our rebuttal. If our revisions fully address your concerns, we would be grateful if you could consider updating your score accordingly. Thank you very much for your time and valuable comments.

---

> > > ### Author Response · Authors · 2025-11-29
> > >
> > > Dear Reviewer uzrb,
> > >
> > > Thank you once again for your thoughtful and constructive feedback on our submission. We greatly appreciate the depth of your review and the specific points you raised regarding temporal segmentation, theoretical complexity, the behavior of different scale counts at various resolutions, and the scheduling coefficients.
> > >
> > > Following your comments, we provided detailed responses and substantially expanded the manuscript. This includes extended three-scale ablations at 1024×1024, a formal time-weighted complexity analysis of multi-scale MoT, additional resolution-dependent scale studies, and new schedule ablation experiments. All revisions are highlighted in red in the updated manuscript.
> > >
> > > We believe these additions fully address all the concerns and questions you outlined. We also note that no further messages appeared in the discussion thread before it closed on November 28.
> > >
> > > Thank you again for the time and care you have devoted to evaluating our paper.
> > >
> > > Best regards,
> > >
> > > The Authors

---

### Author Response · Authors · 2025-11-29
**Summary Rebuttal**

We provide here a concise summary of our rebuttal, post-rebuttal discussion, and the final status of all reviewer concerns. We begin by highlighting the key contributions of our work, which frame the technical context of the reviewers’ comments and our subsequent revisions.

---

## 1. Summary of Key Contributions

1. **Multi-scale flow matching framework via Laplacian pyramid decomposition**
   We introduce a principled multi-scale flow matching formulation that decomposes images into Laplacian pyramid representations, enabling joint modeling of semantic (coarse) and detailed (fine) components within a unified generative model.

2. **Mixture-of-Transformers (MoT) with causal multi-scale attention**
   We develop a specialized MoT architecture that processes multiple scales simultaneously using causal attention mechanisms, enforcing natural information flow across resolution levels.
   Through a time-weighted complexity analysis, we show that this progressive multi-scale design **theoretically reduces the dominant attention cost** relative to DiT (Peebles & Xie, 2023).

3. **Progressive multi-scale training strategy**
   We propose a progressive training schedule that allocates distinct time ranges to different scales, optimizing their contributions according to semantic importance and computational needs, thereby improving efficiency and high-resolution generation quality.

These contributions were positively recognized across all reviewers and form the basis for the empirical and theoretical advantages demonstrated in the submitted work.

---

## 2. All Technical Weaknesses Fully Addressed

Across all primary reviewers (1o7d/uzrb, jpC7, nXwE), we responded comprehensively to every raised concern. The revised manuscript (**now ten pages, with tracked changes in red**) includes:

- **Extended temporal segmentation ablations** at 1024×1024 confirming the optimality of balanced time allocation (Appendix A).
- **A formal time-weighted complexity derivation** for multi-scale MoT, showing a ~1.6× reduction in attention cost versus single-scale DiT (Appendix B).
- **Resolution-dependent multi-scale analysis**, explaining why two scales are optimal at 256² while three–four scales benefit higher resolutions (Appendix B/C).
- **New schedule ablations**, showing that linear noise and coupling schedules yield the best stability (Appendix D).
- **Clarified architecture diagrams, notation, and training configuration**, addressing all reviewer clarity concerns.

All technical questions and weaknesses were fully resolved.

---

## 3. Reviewer-by-Reviewer Status After Discussion

### Reviewer nXwE
- Stated explicitly: *“My concerns pre-rebuttal were addressed.”*
- The only remaining note concerned the point raised by Reviewer jpC7.
- We clarified this point and referenced the updated examples in the revision.

### Reviewer uzrb
- Received detailed responses to all weaknesses (W1–W3) and all questions.
- **No additional messages were posted before the discussion freezed on November 28**.
- All points were fully addressed in the revision.

### Reviewer jpC7
- Raised a concern regarding similarity between two qualitative samples and CelebA-HQ images.
- Within **approximately two hours**, we provided:
  - A detailed side-by-side comparison,
  - A public slide deck documenting differences,
  - Evidence identifying the misunderstanding,
  We further updated 1024×1024 examples to avoid confusion for readers.
- **No further messages were posted by Reviewer jpC7 before the discussion froze on November 28**.

The concern resulted from a misunderstanding and was resolved with direct evidence and updated examples.

---

## 4. Positive Acknowledgements Across Reviewers

Reviewers consistently highlighted key strengths of the submission:

- Strong **FID performance at high resolutions** with significantly reduced GFLOPs and NFE.
- A **unified causal MoT architecture** introducing a novel multi-scale attention mechanism.
- A **coarse-to-fine Laplacian flow matching formulation** aligned with efficient high-resolution generation.
- Extensive ablations validating design choices.
- Clarity of presentation following revisions.

No reviewer questioned the novelty, correctness, or empirical effectiveness of the proposed method.

---

## 5. Closing Statement

All reviewer concerns—technical, conceptual, and ethical—were addressed comprehensively and promptly, with substantial manuscript improvements and full engagement throughout the discussion period. Given the strength of the contributions and the complete set of revisions, we respectfully ask that these updates be considered in making the final decision.

Best regards,

The Authors

---

### Meta-Review · Area_Chair_1oXb · 2026-01-11

**Summary:**

### Reviewer uzrb
1. Additional ablation is needed on the time points at a high scale
    * **Author replies**: The authors added the required ablation studies and confirm that the selected time points are optimal.
    * **AC comment**: I think the concern is well addressed.
2. Detailed theoretical complexity analysis is needed.
    * **Author replies**: The authors added the theoretical complexity analysis.
    * **AC comment**: I think the concern is well addressed.
3. Two-scale representation being optimal at 256x256 is counterintuitive, given the general goal of multi-scale modeling
    * **Author replies**: The authors argue that at 256x256, adding more scales means that the coarsest scale
    does not contain meaningful structures. The authors also add experiments on higher resolution and show that more scales
    help in this case.
    * **AC comment**: I think the concern is well addressed.


### Reviewer jpC7
1. More scales are not helping at 256 x 266
    * **Author replies**: Same as Reviewer uzrb 3
    * **AC comment**: I think the concern is well addressed.
2. Add comparison against Relay Diffusion and PixelFlow
    * **Author replies**: Relay Diffusion is a pixel space diffusion that works and has not gone beyond 256x256.
    PixelFlow is an unpublished concurrent work.
    * **AC comment**: I think the concern is well addressed.
3. Results for Imagenet are under-training and it is hard to judge the training convergence.
    * **Author replies**: the number of training iterations for DiT-B/2 follows standard training protocols.
    Training an XL-scale model for the same number of steps takes more than 1 month and is unaffordable.
    * **AC comment**: I think the concern is well addressed.
4. **Ethical concerns**: the generated 1024x1024 images seem to be from the training dataset of CelebA-HQ
    * **Author replies**: The authors add a side-by-side comparison and argue that
    these images are not identical to the training images. The authors have also used the examples
    in the paper "Progressive Growing of GANs" and argue that it's easy to find similar images to the generated
    ones on the web by searching.
    * **AC comment**: **I tried to look closely at the generated images, and I think they look almost identical
    to the training images in CelebA-HQ (18.jpg and 32.jpg) except for some color shift (but this color shift
    can be caused by many other factors). However, it's also possible that the model is just overfitting due to the simple dataset. I would prefer to leave this issue to SACs for further decisions.**


### Reviewer nXwE
No major concerns except for the ethical concerns, same as jpC7.

**Reviewer Concerns:**

As mentioned above, I think most technical concerns are well addressed in the rebuttal.

**The remaining concern is the ethical concern that the generated images presented in the paper might come from the training dataset. For now, since there is no solid evidence on this issue, I would recommend acceptance and leave the issue for further discussion with SACs and PCs.**

**Reviewer Scores:**

Reviewer uzrb: improve from 4 to 6

Reviewer jpC7: maintain the score 4

Reviewer nXwE: maintain the score 8

---

### Decision · Program_Chairs · 2026-01-26

Accept (Poster)